# LRRC15 mediates an accessory interaction with the SARS-CoV-2 spike protein

**Jarrod Shilts**[1,2], **Thomas W. M. Crozier**[3], **Ana Teixeira-Silva**[3], **Ildar Gabaev**[3], **Pehuén Pereyra Gerber**[3], **Edward J. D. Greenwood**[3], **Samuel James Watson**[3], **Brian M. Ortmann**[3], **Christian M. Gawden-Bone**[3], **Tekle Pauzaite**[3], **Markus Hoffmann**[4,5], **James A. Nathan**[3], **Stefan Pöhlmann**[4,5], **Nicholas J. Matheson**[3,6], **Paul J. Lehner**[3]*, **Gavin J. Wright**[1,2]*

**1** Cell Surface Signalling Laboratory, Wellcome Sanger Institute, Cambridge, United Kingdom, **2** Department of Biology, Hull York Medical School, York Biomedical Research Institute, University of York, York, United Kingdom, **3** Department of Medicine, Cambridge Institute of Therapeutic Immunology and Infectious Disease, University of Cambridge, Cambridge, United Kingdom, **4** Infection Biology Unit, German Primate Center – Leibniz Institute for Primate Research, Göttingen, Germany, **5** Faculty of Biology and Psychology, Georg-August University Göttingen, Göttingen, Germany, **6** NHS Blood and Transplant, Cambridge, United Kingdom

* gavin.wright@york.ac.uk (GJW); pjl30@cam.ac.uk (PJL)

**Data Availability Statement:** Sequencing data are available through the SRA database (accession number PRJNA762706 / https://dataview.ncbi.nlm.nih.gov/object/PRJNA762706?reviewer=

## Abstract

The interactions between Severe Acute Respiratory Syndrome Coronavirus 2 (SARS-CoV-2) and human host factors enable the virus to propagate infections that lead to Coronavirus Disease 2019 (COVID-19). The spike protein is the largest structural component of the virus and mediates interactions essential for infection, including with the primary angiotensin-converting enzyme 2 (ACE2) receptor. We performed two independent cell-based systematic screens to determine whether there are additional proteins by which the spike protein of SARS-CoV-2 can interact with human cells. We discovered that in addition to ACE2, expression of LRRC15 also causes spike protein binding. This interaction is distinct from other known spike attachment mechanisms such as heparan sulfates or lectin receptors. Measurements of orthologous coronavirus spike proteins implied the interaction was functionally restricted to SARS-CoV-2 by accessibility. We localized the interaction to the C-terminus of the S1 domain and showed that LRRC15 shares recognition of the ACE2 receptor binding domain. From analyzing proteomics and single-cell transcriptomics, we identify LRRC15 expression as being common in human lung vasculature cells and fibroblasts. Levels of LRRC15 were greatly elevated by inflammatory signals in the lungs of COVID-19 patients. Although infection assays demonstrated that LRRC15 alone is not sufficient to permit viral entry, we present evidence that it can modulate infection of human cells. This unexpected interaction merits further investigation to determine how SARS-CoV-2 exploits host LRRC15 and whether it could account for any of the distinctive features of COVID-19.

q097hrpl3asgkmhskf71hep8oh). Genome-wide screening data are contained in the Supporting information file.

**Funding:** J.S. and G.J.W. were funded by the Wellcome Trust Grant 206194. J.A.N. was funded by the Wellcome Trust through a Senior Fellowship (215477/Z/19/Z) and by a Pfizer ITEN award. P.J.L. was funded by the Wellcome Trust through a Principal Research Fellowship (210688/Z/18/Z), the MRC (MR/V011561/1), the MRC/NIHR through the UK Coronavirus Immunology Consortium (CiC; MR/V028448/1) the Addenbrooke's Charitable Trust. N.J.M. was funded by the MRC through a Transition Support Fellowship (MR/T032413/1), the Wellcome Trust Institutional Strategic Support Fund (204845/Z/16/Z), NHS Blood and Transplant (WPA15-02) and the Addenbrooke's Charitable Trust. P.J.L. and N.J.M. were supported by the NIHR Cambridge Biomedical Research Centre. The funders had no role in study design, data collection and analysis, decision to publish, or preparation of the manuscript.

**Competing interests:** The authors have declared that no competing interests exist.

**Abbreviations:** ACE2, angiotensin-converting enzyme 2; BSA, bovine serum albumin; COVID-19, Coronavirus Disease 2019; CRISPRa, CRISPR activation; CTD, C-terminal domain; DMEM, Dulbecco Modified Eagle Media; FACS, fluorescence-activated cell sorting; FCS, fetal calf serum; GEO, Gene Expression Omnibus; HBS, HEPES-buffered saline; HBS-T, HBS with 0.1% Tween-20; LRRC15, leucine-rich repeat containing protein 15; MEM, Minimum Essential Medium; Ni-NTA, nickel-nitrilotriacetic acid; NTD, N-terminal domain; PBS, phosphate buffered-saline; RBD, receptor binding domain; RSA, redundant siRNA activity; SARS-CoV-2, Severe Acute Respiratory Syndrome Coronavirus 2; TBS, Tris-buffered saline.

# Introduction

Coronaviruses including Severe Acute Respiratory Syndrome Coronavirus 2 (SARS-CoV-2) have evolved to recognize host proteins through binding interactions that facilitate viral infection. The ongoing Coronavirus Disease 2019 (COVID-19) pandemic caused by SARS-CoV-2 has distinguished itself from past coronavirus outbreaks by its virulence and spread [1], suggesting that the virus has acquired particularly efficient mechanisms to target its hosts [2,3]. Interactions involving the viral spike protein are particularly important to characterize, not only because the spike protein enables cell entry by binding to angiotensin-converting enzyme 2 (ACE2) [4] but also because the spike protein is the central component of many of the current vaccines and therapeutics under development [5,6]. As the most prominent structural feature of the virion surface, it also is positioned to mediate several key stages of viral pathogenesis, and thus its interactions could hold clues to COVID-19 pathology.

While the spike protein is known to interact with human ACE2 receptor for host cell entry, there has been considerable debate around whether ACE2 alone is sufficient to explain COVID-19 pathology [7–10]. For example, ACE2 expression in certain target cells appears to be very low, and other questions remain regarding whether additional factors beyond ACE2 may explain the broad cellular tropism and potent infectivity of SARS-CoV-2 [11–13]. We therefore employed systematic, unbiased screening approaches to identify additional host proteins, which interact with the spike of SARS-CoV-2, and then characterized their activity. Two independent but complementary screening methods identified LRRC15 as a novel host factor inducing binding of the spike protein to human cells by a previously unknown mechanism.

# Results

## Systematic screening for SARS-CoV-2 spike binding interactions identifies LRRC15

While several host proteins have been identified that interact with the spike protein of SARS-CoV-2 based on homology to SARS-CoV-1 [14–16] or heuristics like the C-end rule [17,18], there have been few systematic investigations to determine if other cell surface or transmembrane proteins are targeted by the virus. We therefore carried out two independent large-scale screens in separate laboratories to search for additional cellular interaction targets, which, when expressed, allow human cells to bind spike protein. In the first screen, we arrayed a library of 2,363 full-length human cDNAs encoding most cell surface membrane proteins in the human genome [19] and individually transfected them into HEK293 cells (Fig 1A). Transfected cells in each individual well were stained with fluorescently labeled tetramers of the full SARS-CoV-2 spike extracellular domain, and binding was measured by flow cytometry (S1 Fig and S1 Data). The second screen used a pooled genome-wide CRISPR activation library in RPE1 cells to identify genes, which, when up-regulated, induce the binding of the spike protein S1 domain formatted as an Fc fusion construct (Fig 1B).

Both screens identified *ACE2* as the top-ranked gene as expected, along with other known spike-binding receptors including *CLEC4M* in the arrayed screen (Fig 1C and S2 Data) and the transcription factor *GATA6* in the CRISPR activation screen [20] (Fig 1D and S3 Data). The CRISPR activation (CRISPRa) screen was able to detect even weak spike binding signals, identifying the transcription factor EBF1 despite causing only a small increase in ACE2 levels (S2 Fig and S1 Table and S4 Data). Surprisingly, both screens converged on a gene encoding a membrane protein called leucine-rich repeat containing protein 15 (LRRC15). The top-ranked genes of both screens were individually verified by either rearraying cDNAs individually

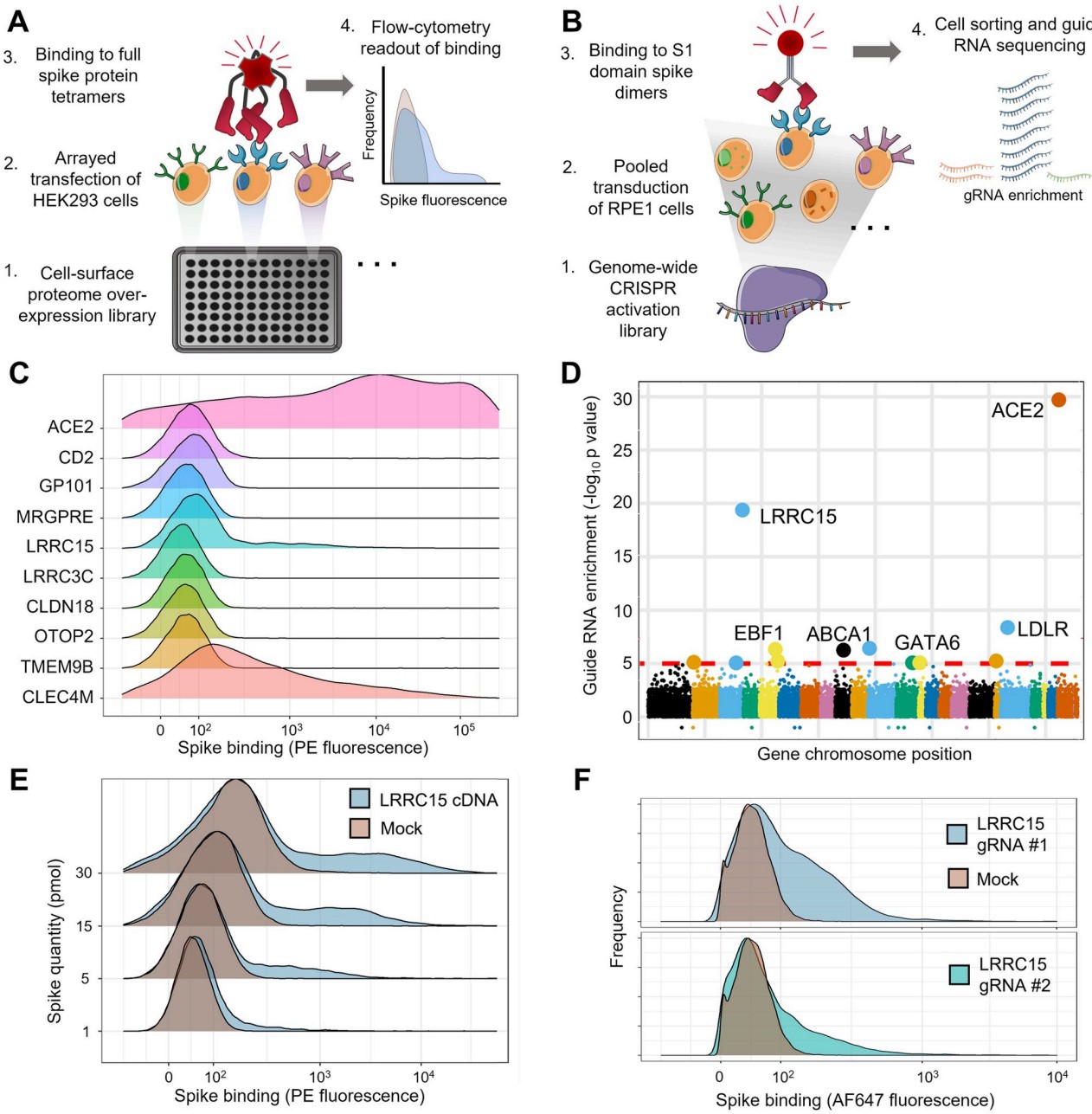

**Fig 1. Arrayed transmembrane protein screening and pooled genome-wide CRISPR activation screening identify LRRC15 as binding SARS-CoV-2 spike protein.** (**A**) Schematic of arrayed cell-based screening to identify host factors that cause SARS-CoV-2 spike protein binding. HEK293 cells were transfected in individual wells of microtiter plates with full-length cDNA constructs encompassing a near-comprehensive library of human membrane proteins. Each well in the array was tested for binding to fluorescent tetramers of full-length SARS-CoV-2 spike by flow cytometry. (**B**) Schematic of pooled CRISPR activation screening. RPE1 cells expressing a SunTag CRISPRa system were transduced with guide RNAs to activate transcription of all genes in the human genome. Cells that bound to Fc protein fusions of the spike S1 domain were sorted by FACS and sequenced to measure guide RNA enrichment post-sorting. (**C**) Top-ranked hits from arrayed cDNA screening. Distributions of cell fluorescence by flow cytometry are shown after incubation with fluorescent spike protein tetramers for cDNAs, which produced top-ranked signals in the arrayed screen. CD2 is included as a negative control. (**D**) Guide RNA enrichment of genes promoting SARS-CoV2 spike protein binding using CRISPR activation screening. Genes are ordered according to their positions across the genome and the statistical significance of their respective guide RNA enrichments post-sorting. (**E**) Independent transfections of LRRC15 cDNA induce spike binding. Flow cytometry traces for HEK293 cells transiently transfected with LRRC15 cDNA compared to mock-transfected controls. Different quantities of full-length spike protein were applied, as indicated along the y-axis. (**F**) Single-guide RNA clones validate *LRRC15* as a gene that induces spike binding. Flow cytometry traces for RPE1 cells transduced and selected for individual LRRC15-activating guide RNAs. CRISPRa, CRISPR activation; LRRC15, leucine-rich repeat containing protein 15; SARS-CoV-2, Severe Acute Respiratory Syndrome Coronavirus 2.

(Fig 1E and S5 Data) or by testing single CRISPR guide RNA clones (Fig 1F and S6 Data). In both cases, LRRC15 was confirmed as a host factor that induces spike binding.

## LRRC15 binding to spike protein is specific and reproducible

After discovering the interaction between the spike protein and LRRC15, we sought to validate and characterize the effect of LRRC15. We were intrigued by the spike protein staining profile of cells transfected with LRRC15, where only a subpopulation of cells gained binding according to a long-tailed distribution. Staining with an antibody against the LRRC15 extracellular domain demonstrated that this profile matched LRRC15 cell surface expression (Fig 2A and S7 Data). Binding was also unambiguous when using preparations of spike protein that were arranged in their more natural trimeric configuration (S3 Fig and S8 Data). Subsequently, we wanted to address two common confounding mechanisms that could account for the cell surface binding. First, we excluded whether LRRC15 is nonspecifically binding to streptavidin or other molecules in our staining reagents, finding that cells overexpressing LRRC15 did not

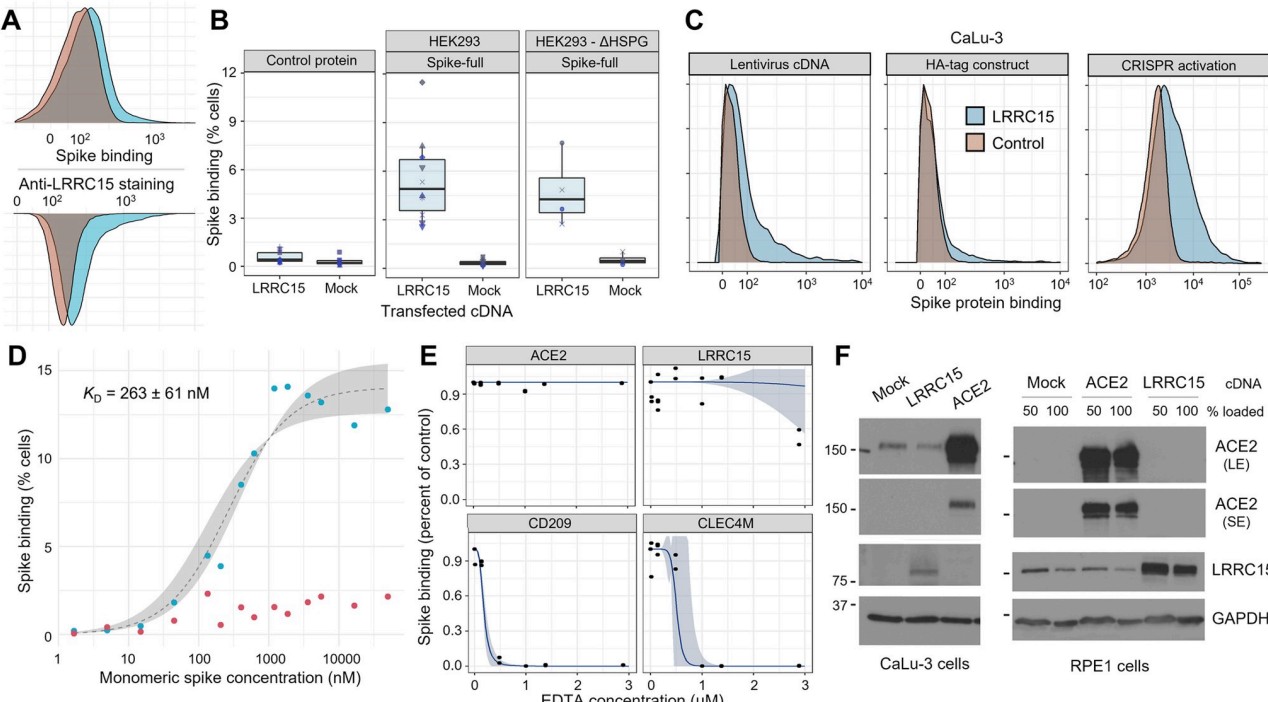

**Fig 2. The spike:LRRC15 interaction is robust to cellular context and differs from previously described spike-binding receptors.** (**A**) Comparison of the cell surface staining profiles of HEK293 cells transfected with LRRC15 cDNA with spike protein (top) and anti-LRRC15 antibody (bottom). (**B**) LRRC15 binding to spike is specific and independent of heparan sulfate proteoglycans. Boxplots summarizing spike binding of LRRC15-transfected cells compared to mock-transfected cells in wild-type HEK293 cells (center panel), a HEK293 strain deficient in cell surface heparan sulfation following SLC35B2 knockout (right panel) and HEK293 cells binding a control tetramer instead of spike (left panel). (**C**) LRRC15 expression consistently induces spike binding. LRRC15 overexpression was achieved in a CaLu-3 human lung cell line using three different approaches and increases in spike binding as quantified by flow cytometry. (**D**) Binding of SARS-CoV-2 spike protein to LRRC15-expressing cells is saturable. Spike binding to LRRC15-expressing (blue) and control (red) cells was quantified by flow cytometry over a wide range of fluorescently conjugated monomeric spike concentrations. A sigmoidal regression curve was fit (gray) to estimate the equilibrium dissociation constant. (**E**) EDTA blocks known lectin receptors for spike protein but does not prevent LRRC15 binding. Dose–response curves (blue) are fit to flow cytometry measurements of spike binding to HEK293 cells under different concentrations of EDTA. (**F**) Expression of LRRC15 is not linked to ACE2 protein production. Western blots with antibodies against human ACE2 and LRRC15 detect no up-regulation of ACE2 when LRRC15 is overexpressed or vice versa in either an ACE2-negative cell line (RPE1) or ACE2-positive cell line (CaLu-3). ACE2 is shown as both short (SE) and long (LE) exposures. Molecular masses are indicated in units of kilodaltons. ACE2, angiotensin-converting enzyme 2; LRRC15, leucine-rich repeat containing protein 15; SARS-CoV-2, Severe Acute Respiratory Syndrome Coronavirus 2.

bind control protein tetramers of Cd4-based linker tags (Fig 2B, left and S9 Data). Second, we tested whether up-regulation of LRRC15 triggers binding through heparan sulfate proteoglycans, which are known to broadly bind many proteins [21,22] including SARS-CoV-2 spike [15]. We observed that genetic ablation of surface heparan sulfates did not affect spike binding to cells presenting LRRC15, indicating little or no contribution of heparan sulfates (Fig 2B, right). We then tested if alternate LRRC15 overexpression methods would also cause a gain of spike protein binding in a more physiologically relevant cell type separate from the initial screens: CaLu-3 lung epithelial cells. Different lentivirally delivered LRRC15 constructs or CRISPR activation again consistently caused cells to gain spike binding (Fig 2C and S10 Data).

To further test the specificity of the interaction, we evaluated two other essential criteria for demonstrating an interaction, which are that the binding be saturable and have sufficient affinity to plausibly occur under physiologic conditions. We covalently conjugated the monomeric spike S1 domain to a fluorophore and measured binding to LRRC15-expressing cells over a wide range of spike concentrations (Fig 2D and S11 Data). We observed binding that was saturable and, by fitting a dissociation curve to the data, we could estimate the monomeric affinity as approximately $260 \pm 60$ nM in terms of its dissociation constant ($p = 0.001$, $t = 4.331$). While weaker than the extremely strong spike–ACE2 interaction, which has a dissociation constant in the range of 1 to 10 nM [4,23,24] (S4 Fig and S12 Data), it is within an order of magnitude of the SARS-CoV-1 spike binding interaction with ACE2 and similar to other known viral receptor interactions [25,26].

## Prior known viral receptors do not account for spike binding upon LRRC15 expression

Although these features appear to distinguish LRRC15 from previously described receptors for SARS-CoV-2, we wanted to investigate the possibility that this apparent interaction was due to LRRC15 overexpression causing another spike-binding receptor to become active. The two principal classes of known spike-binding receptors are the protein ACE2 and C-type lectins (including CD209, CLEC4M, CD207, and ASGR1; [27–29]). NRP1 binding to spike is dependent on proteolytic processing by furin and thus was not expected in our screens [18]. Binding to C-type lectin family receptors is cation-dependent [30,31] and so will be ablated by cation chelating agents such as EDTA. We observed that titrating EDTA at concentrations above those required to block interactions with two control C-type lectin receptors had no effect on the LRRC15 interaction with spike protein (Fig 2E and S13 Data). Similarly, human cells overexpressing LRRC15 had no detectable increase in ACE2 protein (Fig 2F) or transcript levels (S5 Fig and S14 Data), further supporting the distinctness of the LRRC15 interaction.

## LRRC15–spike interaction is distinct to SARS-CoV-2

SARS-CoV-2 is closely related to other coronaviruses including SARS-CoV-1, which was responsible for the SARS epidemic. Although the majority of their basic biology is shared between viruses [32], differences do exist, which are important in accounting for why COVID-19 has reached levels of global morbidity and mortality not seen with previous coronavirus diseases. We asked if this LRRC15 interaction was conserved across other coronaviruses or distinct to SARS-CoV-2. We selected coronavirus lineages that share ACE2 as their primary receptor, including the closely related SARS-CoV-1 betacoronavirus and the more distant NL63 alphacoronavirus [33]. Each of the three respective spike protein orthologs were recombinantly expressed and purified as both full-length ectodomains and as the S1 domain only (Fig 3A). All spike proteins bound ACE2-expressing cells as expected. Surprisingly, the gain in spike binding upon LRRC15 expression was seen only with SARS-CoV-2

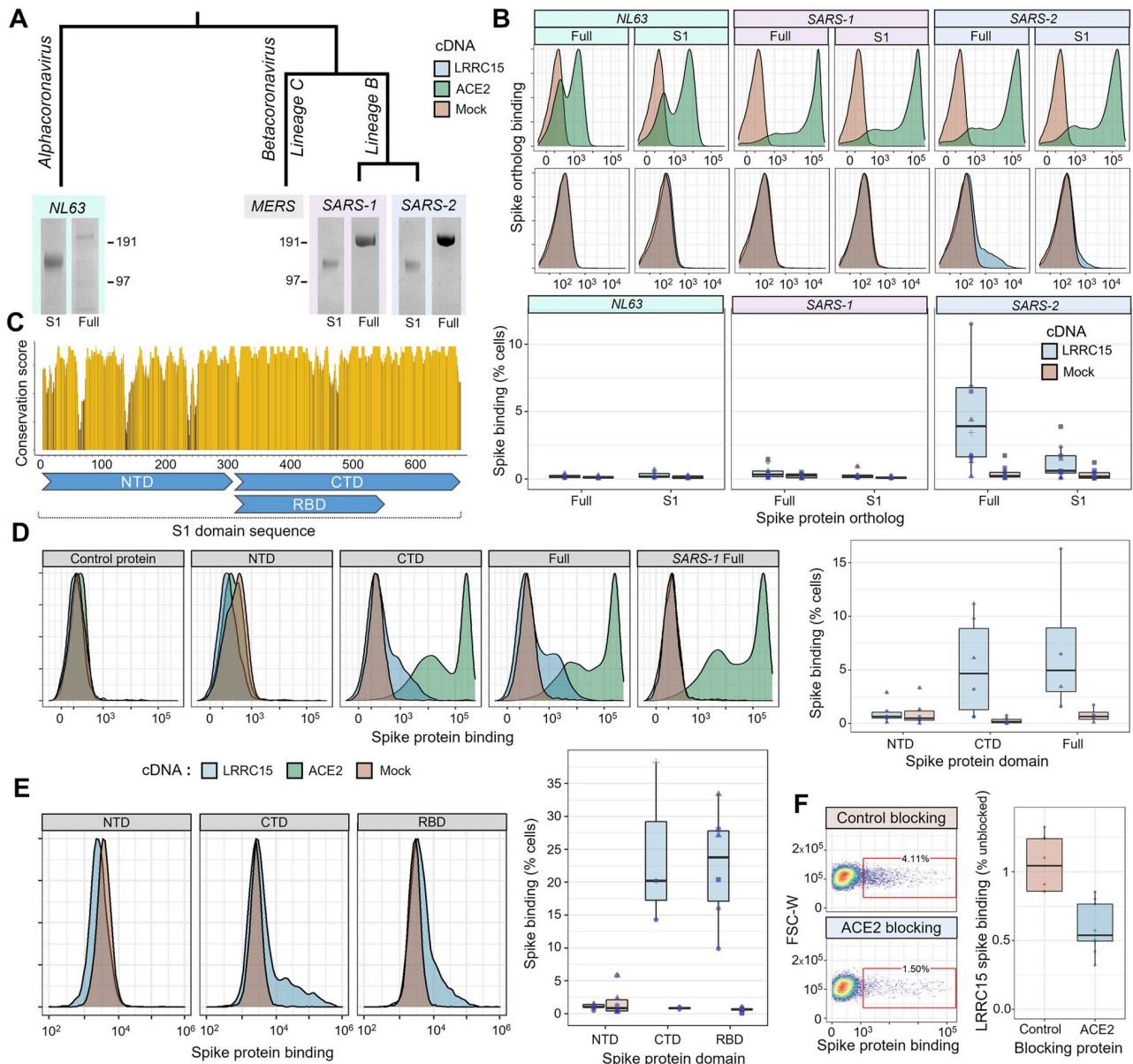

**Fig 3. LRRC15 uniquely interacts with the SARS-CoV-2 spike protein and shares a binding interface with ACE2.** (**A**) Recombinant expression of spike proteins from across the coronavirus family. A labeled phylogenetic tree indicates the relative divergences of coronaviruses above Coomassie-stained gel images of purified recombinant spike proteins. For each virus, the full extracellular domain was produced along with constructs of only the S1 domain. (**B**) Orthologous spike proteins bind ACE2, but only SARS-CoV-2 strongly binds LRRC15. Flow cytometry traces of spike proteins binding to HEK293 cells overexpressing ACE2 (green, top) or LRRC15 (blue, bottom) compared to control cells (red). Quantified replicates for LRRC15 binding are displayed as boxplots below. (**C**) Physical conservation of the amino acids in SARS-CoV-2 spike compared to SARS-CoV-1. The rolling average of physicochemical conservation scores across the spike S1 domain sequence is indicated. Regions corresponding to the NTD, CTD, and RBD are annotated below. (**D**) LRRC15 binding localizes to the spike C-terminal S1 domain. Flow cytometry traces of binding by the CTD but not NTD to LRRC15-transfected cells (left) are shown next to the quantified binding (right). (**E**) The RBD of the SARS-CoV-2 spike protein is sufficient for binding to LRRC15. The percentage of cells binding each spike truncation construct (right) is shown along with representative flow cytometry traces. These data were collected on a different instrument than those shown in the prior panel, accounting for the variation in scale. (**F**) Recombinant ACE2 competitively inhibits spike binding to LRRC15. Dotplots of spike binding by flow cytometry to cells where spike protein was preincubated with a control protein or preincubated with the ACE2 extracellular domain. ACE2, angiotensin-converting enzyme 2; CTD, C-terminal domain; LRRC15, leucine-rich repeat containing protein 15; NTD, N-terminal domain; RBD, receptor binding domain; SARS-CoV-2, Severe Acute Respiratory Syndrome Coronavirus 2.

(Fig 3B and S15 Data). We asked whether this binding pattern may relate to differential glycosylation of the spike proteins, which can shield protein interaction surfaces, and indeed observed that enzymatic deglycosylation of the SARS-CoV-1 spike restored LRRC15 binding (S6 Fig and S16 Data). This suggests the SARS-CoV-1 spike likely has within it a conserved protein interface able to bind LRRC15, but may be less accessible than on the SARS-CoV-2 spike.

## ACE2 and LRRC15 share a binding domain in the spike C-terminal S1 region

A protein sequence alignment between the spike proteins of SARS-CoV-1 and SARS-CoV-2 shows that although they are broadly conserved, there are some notable differences between regions of the protein (Fig 3C and S17 Data). This led us to ask where the binding to LRRC15 localizes. We separately generated SARS-CoV-2 spike constructs of the N-terminal domain (NTD), which has functions related to glycan recognition and presentation [34], and the C-terminal domain (CTD), which includes the ACE2 receptor binding domain (RBD). We observed that all binding was accounted for by the spike CTD (Fig 3D and S18 Data). Within the CTD, binding could be localized specifically to the RBD (Fig 3E and S19 Data). Because ACE2 and LRRC15 shared this binding domain, and the CTD of coronavirus spike proteins are generally known for harboring protein–protein interactions with receptors, we hypothesized that LRRC15 and ACE2 may compete for spike binding. Preincubation of spike with recombinant ACE2 did substantially block LRRC15 binding (Fig 3F and S20 Data, $p < 0.001$, U = 81). These findings suggest a novel function of the RBD of the SARS-CoV-2 spike protein in recognizing host receptors beyond ACE2.

## LRRC15 is found in mesenchymal and endothelial cells within tissues where ACE2 is present

After accumulating biochemical evidence for this novel interaction, we next sought to understand its potential role in COVID-19 pathology. There is little prior literature on LRRC15, so we began by establishing where in the human body this protein is present, and how that distribution matches with the known tropism of SARS-CoV-2. We examined a detailed whole-body proteomic atlas [35] for evidence of LRRC15 expression. We compared the LRRC15 tissue distribution to ACE2, which, as the viruses' primary receptor, should mark which tissues are virally susceptible (Fig 4A and S21 Data). Despite emerging independently from a genome-wide screen, LRRC15 shared similarities to the tissue distribution of ACE2, with major targets of viral infection such as the lungs and gastrointestinal tract also showing the most abundant expression of LRRC15 (Fig 4B and S22 Data).

Because lung tissue is the main source of severe COVID-19 symptoms and expresses among the highest levels of LRRC15 in the body, we conducted a more detailed analysis of *LRRC15* expression within the lungs using public single-cell transcriptomic datasets [36–38]. We began by considering reference data from healthy uninfected lungs. Although datasets of the upper respiratory tract did not report *LRRC15* either because of lower sensitivity or a true absence [39,40], in the lower respiratory tract it was readily detected, predominantly in mesenchymal cells and endothelium (Fig 4C and S23 Data). The overlap with cell types expressing *ACE2* was relatively weak, although as previously reported, these methods only detect *ACE2* at very low levels and incompletely [11,41,42].

Severe COVID-19 is associated with substantial inflammation and other changes to the lungs, which could influence how LRRC15 is distributed as infection progresses. When we analyzed gene expression datasets of lung fibroblast cells stimulated with the cytokine

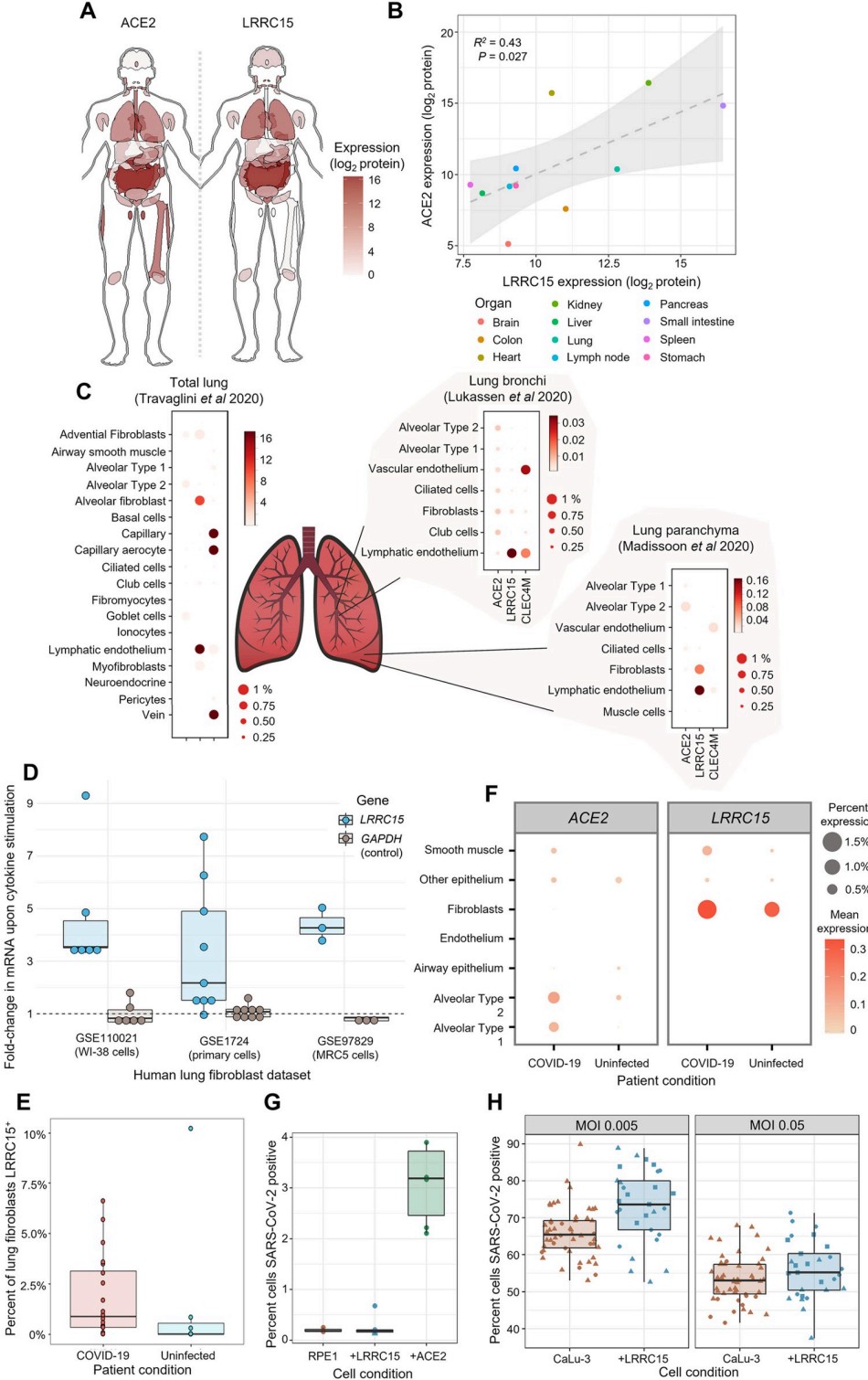

**Fig 4. LRRC15 is expressed in virus-susceptible tissues and can modulate SARS-CoV-2 infection.** (**A**) Tissue distribution of LRRC15 and ACE2 expression based on a whole-body proteomic atlas [35]. For each tissue, protein expression is reflected by the red color intensity scale provided. (**B**) Organs with high ACE2 expression tend to also express substantial LRRC15. Each data point represents protein abundance in a major human organ or tissue as measured by mass spectrometry. A linear regression line and 95% compatibility interval is shaded in gray. (**C**) Single-

cell transcriptome measurements of human lung specimens identify cell type distributions for LRRC15 expression. The percentage of cells where a gene transcript was detected is indicated by size, while average counts per cell are color-shaded. Expression of spike receptors *ACE2* and *CLEC4M* are shown for comparison. (**D**) Inflammatory cytokines greatly up-regulate the expression of *LRRC15* in lung fibroblasts. Three different publicly available datasets are shown that exposed cultures of human lung fibroblast cells to TGF-beta. Relative changes in mRNA expression were calculated for *LRRC15* and, as a negative control not expected to change, *GADPH*. (**E**) COVID-19 patients have greater proportions of *LRRC15*-expressing fibroblasts in their lungs compared to uninfected patient controls. Each point is a different lung tissue donor measured in a recent single-cell RNA-seq study [49] showing the percentage of single fibroblast cells where *LRRC15* mRNA was detected. (**F**) The lungs of COVID-19 patients experience a broad expansion in both *LRRC15* and *ACE2* expressing cells. For each cell population identified by single-cell RNA-seq [49], the expression (as measured both by average mRNA counts and percentages of cells) was compared between deceased COVID-19 patients and uninfected controls. (**G**) Expression of LRRC15 is insufficient to make cells permissive to SARS-CoV-2 infection. LRRC15 or ACE2 were overexpressed by transducing RPE1 cells with CRISPRa sgRNA followed by infection with a recombinant SARS-CoV-2-ZsGreen reporter virus. (**H**) LRRC15 overexpression changes the susceptibility of CaLu-3 lung cells to viral infection. CaLu-3 cells were transduced with lentiviruses to overexpress LRRC15 followed by infection with SARS-CoV-2-ZsGreen. Viral infection was quantified by counting GFP-positive cells by fluorescent microscopy in (D) and (E). ACE2, angiotensin-converting enzyme 2; COVID-19, Coronavirus Disease 2019; CRISPRa, CRISPR activation; LRRC15, leucine-rich repeat containing protein 15; SARS-CoV-2, Severe Acute Respiratory Syndrome Coronavirus 2.

TGF-beta, we found dramatic up-regulation of *LRRC15* expression (Fig 4D, $p < 0.001$, $t = 5.7$; S24 Data). This translated to noticeable expansions of *LRRC15*-expressing fibroblasts within the lungs of patients with severe COVID-19 (Fig 4E, $p = 0.06$, $U = 103$; S25 Data). More broadly, there were increases in both *LRRC15* and *ACE2* levels in COVID-19 patients, suggesting that a common response pathway may expand the role of both during infection (Fig 4F and S26 Data). Interestingly, the relative levels of LRRC15 in COVID-19 patients have been reported to correlate to a patient's viral burden [43], implying that this induction of *LRRC15* expression may be consequential to infection.

## Cell surface LRRC15 is not sufficient to make cells permissive to SARS-CoV-2 infection

Because there has been only relatively limited prior characterization of LRRC15, it raised ambiguities as to whether it can appear on the surface of cells that express it, or if it acts intracellularly and thus more likely to be involved in stages following viral entry [44,45]. We initially tested whether cell surface LRRC15 could act as an entry receptor for the virus, following our observation that LRRC15 predominantly localizes to the cell surface of our cell line models (S7 Fig). We challenged RPE1 cells with a recombinant SARS-CoV-2-ZsGreen reporter virus [46]. RPE1 cells are not naturally permissive to infection but become infectable upon introduction of ACE2. However, expression of LRRC15 was insufficient to make the cells permissive to infection (Fig 4G and S27 Data).

## Presence of LRRC15 modulates the ability of SARS-CoV-2 to infect host cells

Direct entry receptors comprise only one type of host–virus interaction. We next sought to gain a broader view of what functional role LRRC15 has in SARS-CoV-2 infection. In order to examine whether LRRC15 might modulate the efficiency of viral entry into ACE2-expressing cells, we employed the naturally susceptible human lung cell line (CaLu-3) that is commonly used as a model for coronavirus infection [47]. We introduced LRRC15 ectopically to these cells to emulate the expression seen in other human lung cell populations we identified in our expression analysis. Compared to unmodified CaLu-3 cells that lack any detectable LRRC15,

the expression of LRRC15 modestly enhanced viral infection (Fig 4H, MOI 0.005: $p$ = 0.0008, U = 396; MOI 0.05: $p$ = 0.14, U = 580; S28 Data). At high viral titers, the effect of LRRC15 was dampened, which we interpret as indicating that when virus is abundant then sufficient virions will encounter ACE2 receptors for entry without the ancillary action of LRRC15. We also investigated the alternative scenario where LRRC15 and ACE2 are expressed on different cells, to measure whether we could detect a *trans* effect on infection. Coculturing ACE2-expressing cells in the presence of cells overexpressing LRRC15 did not appreciably enhance nor inhibit infection of the ACE2-expressing population when using wild-type isolates of SARS-CoV-2 (S8 Fig and S29 Data). Similarly, when we tested a sequential model where virions first encounter LRRC15-expressing cells and then subsequently are exposed to ACE2-expressing cells, we likewise did not observe evidence for a strong *trans* effect on infection (S9 Fig and S30 Data). These results point to a mechanism where LRRC15 acts predominantly as a modulator of infectivity for already-susceptible host cells.

## Discussion

In this report, we describe LRRC15 as a novel interaction target for the SARS-CoV-2 spike protein. We demonstrate that the binding differs from previously known classes of spike binding receptors and is distinct to SARS-CoV-2. Although cell surface LRRC15 appears to be insufficient to permit viral entry, we present initial evidence that LRRC15 could modulate a host cell's susceptibility to SARS-CoV-2. Our analysis of public expression datasets suggests LRRC15 is expressed in tissues susceptible to infection at levels comparable to or exceeding those of ACE2. In the lung, these include several cell types that have been detected as virally infected targets in human patient samples [48–53], although, interestingly, the highest LRRC15 levels are not associated with the alveolar epithelial cell types that are most commonly studied. These fibroblast and endothelial populations where LRRC15 is most abundant have clear significance to COVID-19 pathology [49,53–55], but it will be important to determine how LRRC15 may be involved in disease pathways where these cell types are associated. In our study, we focused on describing the lungs, but given the coexpression we find between LRRC15 and ACE2 in other tissues, a related interesting question is whether cells outside of the lung can reveal where LRRC15 expression is clinically relevant. For example, case reports of SARS-CoV-2 infection in the brain implicate lymphatic vessel cells where LRRC15 is particularly plentiful [56].

The normal molecular function of LRRC15 in the human body is not well characterized [44,57,58]. The protein's 15 leucine-rich repeat domains in its extracellular region suggest that it may engage in protein–protein interactions (analogous to LRRC15's closest human paralog, the platelet receptor member GP5), although it contains no obvious cytoplasmic signaling motifs [59]. Similarly, it may also have physical interactions with components of the extracellular matrix [60]. Although LRRC15 has been found by us and previous researchers to be abundant on the cell surface, some quantity appears present in intracellular membranes, and these could conceivably be acting through other pathways beneficial to the virus' proliferation such as via directing protein trafficking [61]. Based on our inability to recombinantly produce a functional version of LRRC15 when truncating its extracellular domain, we conjecture that the transmembrane domain or cytoplasmic cysteine-rich region of the protein may also be required. A recent study on LRRC15 similarly found the presence of a membrane-spanning domain was essential for binding activity to be observed [62]. However, our data are also compatible with the possibility that the interaction be indirect yet distinct from other previously known interaction pathways. Despite its lack of biochemical characterization, LRRC15 is remarkably well conserved among mammals, including being approximately 90% identical in

sequence between human and nearly all proposed host species for SARS-CoV-2 (S10 Fig and S31 Data). Elucidating what function LRRC15 has evolved for within these hosts could provide a critical clue to determining how this interaction acts during viral infection.

Notably, two studies of COVID-19 patient cohorts independently uncovered associations with LRRC15. In an analysis of lung tissue samples to find transcriptional signatures of fatal COVID-19, *LRRC15* was found to be significantly higher in fatal COVID-19 cases compared to controls [63]. While the direction of causality in that first study was ambiguous, a second group conducted a longitudinal study of high-risk COVID-19 patients to monitor proteins detectable in serum that predict whether a COVID-19 patient will go on to develop severe or only mild disease [64]. They found LRRC15 as their single strongest predictor of patient's outcomes. Seeing these separate analyses of COVID-19 patients converge on the same host factor we identified through functional screening further emphasizes the value of closer investigation into LRRC15 during SARS-CoV-2 infection, now that our work adds a potential biochemical mechanism to explain clinical observations.

After we first made our results publicly available as a preprint, two independent research groups subsequently reported very similar results to our own in separate preprints, including identifying LRRC15 as the only novel host protein to be discovered after genome-wide screening, localizing the RBD as the interaction interface, determining that LRRC15 can modulate infection without itself acting as an entry receptor for SARS-CoV-2, and analyses of the expression patterns of LRRC15 in the human body suggesting links to COVID-19 [65,66]. These studies not only provide independent replication for our core findings but also offer several analyses that extend our original findings. While there is agreement that LRRC15 can have an enhancing role when present on cells expressing ACE2 [66], whether LRRC15 also has an inhibitory effect on infection in *trans* is one point of divergence between these studies. Our own data have not detected any *trans* effect that LRRC15-expressing cells have on the infection of nearby susceptible target cells. This suggests that this particular phenotype maybe observed only in particular experimental contexts (relative cell densities, receptor expression levels, et cetera). Future work aimed at understanding the physiological conditions of the lung microenvironment can resolve the relative contributions of these proposed mechanisms to natural infections.

Because of the wide clinical significance of the SARS-CoV-2 spike protein, awareness of this novel interaction has other implications. Several classes of successful therapeutics operate by blocking spike protein interactions with human host receptors, most notably in the form of ACE2-blocking biologics [67–69]. Our identification of a shared binding domain between LRRC15 and ACE2 implies many of these may also prevent spike–LRRC15 interactions, at least extracellularly. The development of drugs targeting LRRC15 has already been initiated by oncologists who observed elevated LRRC15 in certain tumors [58,70]. These could provide a means for further dissecting the biology of LRRC15 in infection and eventually may be possible to repurpose in patient settings.

These findings underscore the significance of the discovery of LRRC15 as a novel host factor for SARS-CoV-2. Although our investigations so far have only just begun to reveal what roles LRRC15 may have in COVID-19, there is considerable potential for dissecting how LRRC15 acts during infection that builds off the molecular binding mechanism we elucidate here. The promising initial data reported by ourselves and others support that LRRC15 could be a useful target for therapeutics and may be informative when addressing important unresolved questions from the COVID-19 pandemic about what makes SARS-CoV-2 so distinctly pathogenic and what explains why patient outcomes are so variable.

## Materials and methods

### HEK293 cell culturing

For both binding assays and the production of recombinant proteins, suspension HEK293-E cells [71] were grown in Freestyle media (Gibco #12338018) supplemented with 1% (m/v) fetal bovine serum (HyClone #SH30071) at 37°C in 5% $CO_2$ and 70% humidity. Cells were transiently transfected 24 hours after being split to a density of $2.5 \times 10^5$ cells per mL [13]. For protein production, transfections were performed with 0.5 μg plasmid DNA per mL cells, while for cDNA expression, 1 μg per mL was used. When expressing recombinant proteins that were to be covalently biotinylated, cells were also cotransfected with 40 ng per mL of a BirA expression plasmid [72] and the cell media was supplemented with an additional 100 μM D-biotin (Sigma #2031). For binding assays, cells were incubated a further 40 to 48 hours after transfection before being used in experiments. For protein production, the incubation time was 96 to 120 hours.

### Recombinant protein purification

After HEK293 cells were transfected and allowed to incubate, they were centrifuged at 2,000 × *g* for 20 minutes. Recombinant protein constructs all contained signal peptide secretion sequences, and, hence, the supernatant fraction was collected for purification. For affinity purification to the protein's His-tags, nickel-nitrilotriacetic acid (Ni-NTA) resin (Thermo Scientific #88221) was prepared with two washes for 10 minutes each in 25 mM imidazole (Sigma #I2399) phosphate buffer. Supernatant filtered through 0.22 μm filters was then mixed with the Ni-NTA resin and incubated with agitation overnight at 4°C. Samples were washed three times with 25 mM imidazole phosphate buffer (Sigma #I2399), incubating 5 minutes in between washes. Finally, proteins were eluted using 200 mM imidazole buffer, and resin was separated on polypropylene columns (Qiagen #34924). SARS-CoV-2 S1-Fc protein was purified from serum-free supernatant of transfected 293T cells by Protein A affinity chromatography using hiTrap MabSelect PrismA columns (GE Healthcare) and the Äkta-Pure liquid chromatography system. Experimental replicates all feature different batches of purified protein.

### Protein construct designs

SARS-CoV-2 spike protein constructs were taken from previously published designs [13,16]. The full extracellular domain (Q14-K1211) was mutated to remove its polybasic cleavage site (682–685 RRAR to SGAG) and introduce a proline stabilizing mutation (986–987 KV to PP). S1 domain truncations were made at Y674. Orthologous spikes were aligned to SARS-CoV-2 to define similar boundaries. These for SARS-CoV-1 were at S14-L666 (S1 domain) and S14-K1193 (full extracellular), and for NL63 at C19-V745 (S1 domain) and C19-K1294 (full extracellular). The boundary for the SARS-CoV-2 NTD and CTD was at the junction between F318 and R319 within the S1 domain [23]. The RBD construct spanned R319—F541. To produce trimer spike protein, we fused the T4 foldon trimerization domain to the C-terminus as previously reported [4,24]. Recombinant ACE2 for blocking experiments had its extracellular domain defined as M1-S740. All recombinant proteins contained a tag and linker region described previously [73,74] that includes a 6-His tag for purification, fragment of rat Cd4 for stabilization, and a biotin acceptor peptide for enzymatic monobiotinylation. Protein control constructs consisted solely of this tag region. cDNA constructs for LRRC15 corresponded to the complete sequence (NM_001135057.2) derived from a commercial cDNA library (Origene

#SC325217). cDNA for ACE2 (NM_021804.2) was derived from a similar expression plasmid (Geneocopia #EX-U1285-M02). All nucleotide sequences were verified by Sanger sequencing.

## Tetramer normalization ELISAs

To calculate optimal stoichiometries for forming recombinant protein tetramers, competitive ELISAs were performed where biotinylated proteins were titrated against streptavidin. In 175 μL HEPES-buffered saline (HBS) with 0.1% Tween-20 (HBS-T, Sigma #P2287), 96-well streptavidin-coated plates (Greiner #655990) were rinsed, then blocked for 1 hour in 2% (m/v) bovine serum albumin (BSA, Sigma #A9647) in HBS. In a separate polystyrene 96-well plate (Greiner #650161), a 2× dilution series of the biotinylated recombinant protein was made in a solution of 2% BSA HBS. To each well of the dilution series, 1.5 pmol of fluorescently conjugated streptavidin (Biolegend #405245 or Biolegend #405237) was added and allowed to incubate for at least 1 hour. Samples were then transferred to the blocked streptavidin-coated plate. Free biotinylated proteins were then allowed to be captured by the plate over at least 45 minutes, before three washes with 150 μL HBS-T. For primary antibody staining, 1.6 μg/mL of a monoclonal OX68 antibody against the recombinant protein tag region was incubated for 1 hour. Following three more HBS-T washes, 0.2 μg/mL of a secondary anti-mouse IgG antibody linked to alkaline phosphatase (Sigma #A9316) was incubated for 30 minutes. Plates were washed a final three times with HBS-T before 60 μL substrate was added in the form of 2 mg/mL para-Nitrophenylphosphate (Sigma #P4744) in diethanolamine buffer. The reaction was allowed to proceed for approximately 30 minutes before absorbance at 405 nm was measured on a Tecan Spark plate reader. The minimum concentration of biotinylated protein at which signal was reduced to baseline (indicating stoichiometric equivalence with the 1.5 pmol streptavidin) was selected for assembling tetramer reagents.

## Cytometry binding assays

Human cell lines transiently transfected or lentivirus-transduced to overexpress LRRC15 (or mock-treated) were transferred into 96-well u-bottom plates (Greiner #650161) at approximately $5 \times 10^4$ cells per well. To 50 μL cells, 50 μL of DAPI was added for a final concentration of 1 μM DAPI. Plates were then incubated on ice for 5 minutes. To remove the supernatant, plates were centrifuged at 4°C for 3 minutes at $200 \times g$. Cells were resuspended in 100 μL solutions of preconjugated recombinant protein in 1% BSA in phosphate buffered-saline (PBS) supplemented with calcium and magnesium ions (Gibco #14040133). To assemble recombinant protein tetramers, a fixed quantity of streptavidin linked to R-phycoerythrin (Biolegend #405245) or Alexa Fluor 647 (Biolegend #405237), typically 15 pmol or 30 pmol, was allowed to incubate for at least 1 hour with the covalently biotinylated recombinant proteins at the concentration empirically determined by the tetramer normalization ELISA (see above). For recombinant Fc fusion proteins, purified spike protein was directly added to a final concentration of 16 μg/mL. Cells were left to bind the recombinant proteins over at least 45 minutes on ice. To wash, an additional 100 μL of PBS was added, plates centrifuged again to decant supernatant, then resuspended in 200 μL PBS before another centrifugation. In the case of fluorescent-conjugated tetramer staining, cells were finally resuspended in 1% BSA in PBS, while in the case of Fc fusion protein staining, cells were stained with secondary antibody conjugated to Alexa Fluor 647 against human IgG-gamma1 (Jackson #109-605-006), washed, and then resuspended. Cells were measured on either a BD LSR Fortessa or CytoFLEX LX flow cytometer. The gating strategy is illustrated in the Supporting information (S11 Fig).

## Arrayed cDNA screen

The human cDNA library used for the arrayed screen was previously described [19] and covers the vast majority of human cell surface proteins [75–77]. The screening procedure followed the "Cytometry Binding Assays" protocol above, with the following modifications. After 24 hours of HEK293 cells incubating in a standard shaking flask (Corning #431143), the cells were transferred to u-bottom 96-well plates for transfections. In each well, 100 μL of cells at a density of $5 \times 10^5$ cells/mL were then transfected with 200 ng cDNA plasmid. Approximately 46 hours after transfection, cells were stained with full-length SARS-CoV-2 spike tetramers around streptavidin R-phycoerythrin (Biolegend #405245) as described above, except only a 5 pmol quantity of protein was used. To avoid screen hits unrelated to protein–protein interactions due to heparan sulfate binding, we used a previously described HEK293 cell line with a biallelic targeted disruption of the *SLC35B2* sulfate transporter, which prevents cell surface heparan sulfate presentation [22,78,79]. The gate for positive protein staining was set at a fluorescence value of $10^3$ across all plates. All hits from the arrayed screen were repeated individually using the standard procedure described above, which resolved false-positives caused by occasional blockages to the laminar flow of the cytometry instrument when running such a large quantity of samples.

## CRISPRa screen

A genome-wide CRISPRa screen was performed using a clonal RPE-1 cell line containing SunTag CRISPRa system (a gift from the Tanenbaum lab), where a nuclease-dead Cas9 (dCas9) is fused to 10xGCN4 peptide array (with a P2A-mCherry reporter) together with a GCN4 nanobody fused to GFP and the transactivator VP64. A total of $1.26 \times 10^8$ cells were transduced with CRISPRa sgRNA library (Addgene #1000000091) lentivirus at an MOI of 0.3 (180-fold coverage), which was assessed by flow cytometry (BFP$^+$) at 72 hours post-infection. sgRNA containing cells were enriched by selecting with puromycin (10 μg/mL) up until the first sort. Cells were harvested for fluorescence-activated cell sorting (FACS) at day 10 post-transduction. Trypsinized cells were blocked with rabbit IgG (EMD Millipore, 20 μg/mL) for 10 minutes in sort buffer (PBS + 2% FCS), to which purified S1-Fc (final concentration 16 μg/mL) was added and incubated for 40 minutes on ice. Cells were then washed in sort buffer and stained with an AF647 conjugated F(ab′)$_2$ antibody against human IgG-gamma1 (Jackson #109-605-006, 1.5 μg/mL) for 30 minutes on ice. A total of $1 \times 10^8$ cells (90% BFP$^+$) were sorted on BD-Influx cell sorters. The top 2.8% most AF647$^+$ population, amounting to 2.6 x $10^6$ cells, were collected and used directly for DNA extraction (Qiagen, Gentra Puregene). Cells were gated to ensure mCherry and GFP reporters did not change and sgRNA expression was selected by sorting for BFP$^+$ cells. An unsorted library population was maintained separately at 180-fold coverage throughout the experiment, and DNA from this population was also extracted at day 10. sgRNA sequences were amplified and Illumina sequencing adaptors added by two sequential rounds of PCR followed by PCR purification (AMPure XP, Beckman Coulter). Next-generation sequencing was performed on a MiniSeq System (Illumina) using a custom primer. For data analysis, single-end 35 bp reads were trimmed down to the variable sgRNA segment using FASTX-Toolkit and aligned to an index of all sequences in the library using Bowtie 2. Read count statistics were generated using the RSA algorithm [80]. Sequencing data have been submitted to SRA and are available under accession PRJNA762706.

## Antibody staining

For antibody staining for flow cytometry, HEK293 cells were incubated with polyclonal against the LRRC15 extracellular domain (LSBio #LS-C165855) in a 1:50 dilution in 1% BSA PBS.

Staining was done in u-bottom 96-well plates (Greiner #650161) for 30 minutes on ice. Plates were then centrifuged at $200 \times g$ for 3 minutes to remove the supernatant and washed twice in PBS. A 1:800 dilution of anti-rabbit IgG secondary antibody conjugated to Alexa Fluor 488 (Jackson #111-547-003) in 1% BSA PBS was incubated for 20 minutes on ice. One additional PBS wash was performed before resuspending the cells in 1% BSA PBS and measuring them on a BD LSR Fortessa flow cytometer. For immunofluorescence staining, CaLu-3 were plated on 13 mm, round, #1.5 coverslips (VWR) and allowed to adhere for 72 hours. Media was exchanged for methanol-free 4% PFA in PBS and incubated for 15 minutes. Coverslips were washed three times in Tris-buffered saline (TBS) (pH 7.4) and subsequently permeabilized in 0.1% Triton X-100 in TBS for 5 minutes before being blocked with 2% BSA (Arcos) in TBS for 20 minutes. Coverslips were then incubated in TBS + 2% BSA containing the indicated dilution of primary rabbit anti-human LRRC15 antibody (ab150376) for 1 hour at room temperature, washed three times in TBS + 2% BSA, incubated for 30 minutes in TBS + 2% BSA containing highly cross-adsorbed goat anti-rabbit Alexa Fluor 555 conjugated secondary antibody (2 μg/mL) and DAPI (0.2 μg/mL), then washed three times in TBS and once in distilled water prior to mounting slides with Prolong Glass (Thermo Fisher) followed by overnight curing. Samples were observed on a Zeiss 980 equipped with an Airyscan2 using the 405 nm (DAPI) and 555 (Alexa Fluor 555) lasers in Airyscan mode to provide high-resolution images.

## Affinity measurement assay

Cell-binding assays to calculate monomeric affinity were modified versions of "Cytometry Cell Binding" protocol above. Recombinant SARS-CoV-2 spike protein S1 C-terminal domain was dialyzed into PBS (Millipore #71505) and then covalently conjugated to R-phycoerythrin using a commercial amine coupling kit (Abcam #ab102918). Transfected HEK293 cells were transferred to v-bottom 96-well plates (Greiner #651261). Comparatively small quantities of cells were used (10 μL, or about a few thousand cells per well) in order to ensure that the assumptions around free ligand concentration for the 1:1 binding model equation were not violated [81]. After the initial wash and DAPI stain, cells were resuspended in a 3× dilution series of spike protein going across the wells of the plate, starting at 16.7 μM diluted in 1% BSA in PBS with calcium and magnesium ions (Gibco #14040133). Binding was allowed to reach equilibrium over an hour at 4°C before two washes in PBS. For washes, cells were centrifuged at $300 \times g$ for 7 minutes at 4°C, then supernatant was carefully aspirated by pipette. Cells were finally resuspended in 40 μL 1% BSA PBS and measured with a BD LSR flow cytometer.

## Coomassie protein staining

Purified protein samples were characterized by Coomassie total protein staining. Proteins were first denatured in NuPAGE sample buffer (Invitrogen #NP0007 and #NP0004) by heating to 70°C for 10 minutes. These were then loaded on to 4% to 12% gradient Bis–Tris gels (Invitrogen #NP0329) and electrophoresed for 50 minutes at 200 volts. Gels were removed from their cassettes, briefly rinsed in pure water, and then stained with Coomassie R-250 (Abcam #ISB1L) overnight at room temperature. After a brief rinse in water, gels were imaged under visible light filters with an Azure c600 system.

## Western blotting

Cells were collected by trypsinization and washed 3 times with PBS ($1,000 \times g$, 5 minutes, 4°C). Cell pellets were resuspended in lysis buffer (1% (w/v) digitonin, 1× Roche cOmplete protease inhibitor, 0.5 mM PMSF, 10 mM Tris–HCL (pH 7.4)) and incubated on ice for 40 minutes. The lysates were then centrifuged ($17,000 \times g$, 10 minutes, 4°C) and the post-nuclear

fractions were transferred to new tubes. The protein concentration of each sample was determined by Bradford assay. Samples were adjusted with TBS buffer and 6× Laemmli buffer + 100 mM dithiothreitol (DTT) and heated at 50˚C for 10 minutes. Samples were separated by SDS-PAGE and transferred to PVDF membranes (Merck), then blocked in 5% milk + PBS-T (PBS + 0.2% (v/v) Tween-20) for 1 hour. Blocked membranes were incubated with primary antibody (LRRC15 [LSBio aa393-422], ACE2 [Abcam Ab108252], GAPDH [GeneTex GTX627408]) in 5% milk + PBST (PBS + 0.2% (v/v) Tween-20) at 4˚C overnight, then incubated with peroxidase (HRP)-conjugated secondary antibodies (Peroxidase AffiniPure Goat Anti-Rabbit IgG (H+L) [Jackson ImmunoResearch 111-035-144], Peroxidase AffiniPure Goat Anti-Mouse IgG (H+L) [Jackson ImmunoResearch 115-035-146]) for 90 minutes at room temperature.

## qPCR

Total RNA was extracted using the RNeasy Plus minikit (Qiagen) following the manufacturer's instructions and then reverse transcribed using Protoscript II Reverse Transcriptase (NEB). Template cDNA (20 ng) was amplified using the ABI 7900HT Real-Time PCR system (Applied Biotechnology or Quantstudio 7, Thermo Scientific). Transcript levels of genes were normalized to a reference index of a housekeeping gene (β-actin).

## Phylogenetics analysis

Homologs for LRRC15 were identified by BLAST searches against the human LRRC15 amino acid sequence, filtering for species reported to be potential SARS-CoV-2 hosts [82,83]. Multiple sequence alignments were done by the Clustal Omega algorithm using default parameters [84]. Phylogenetic trees were drawn using Jalview (v. 2.11.1.4) from average distance calculations based on the BLOSUM62 substitution matrix. For coronavirus spike protein comparisons, SARS-CoV-1 spike and SARS-CoV-2 spike protein sequences were pairwise-aligned, and conservation at each amino acid position was calculated based on an established physicochemical conservation score [85]. Measurements for the alpha and beta coronavirus phylogenetic tree were taken from a prior study on whole-genome nucleotide sequences [33].

## Expression data analysis

Whole-body proteomics data were downloaded from a prior study [35]. The data were visualized with the gganatogram pacakge (v. 1.1.1) in R (v. 4.0.3). An expression value of at least 500 molecules (based on intensity-based absolute quantification; [86]) was required for a protein to be displayed as expressed in a given tissue. Healthy lung single-cell RNA sequencing datasets were downloaded from the COVID-19 Cell Atlas and related resources [36–38,41]. Paired datasets of COVID-19 patient lungs and matched controls were downloaded from a prior study [49]. Data from human lung fibroblasts stimulated by TGF-beta were identified through the Gene Expression Omnibus (GEO) and RNA-seq expression values from the original studies [87–89] were converted to relative fold-changes for the purposes of cross-study comparison. The data were visualized with the Scanpy package (v. 1.4.5) in Python (v. 3.7.4). Datasets with no detected read counts for either ACE2 or LRRC15 were excluded.

## Statistical calculations

Whenever two groups were compared, a Mann–Whitney U test was used to compare distributions without invoking normality assumptions. Redundant siRNA activity (RSA) analysis was used to calculate p-values to identify statistically significantly enriched genes, comparing read

counts from individual sgRNAs from harvested sorted cells to an unsorted library population [80]. The statistical significance of the dissociation constant fit in the monomeric binding affinity model was computed using the *nls* function in the R stats package (v. 4.0.3). Confidence intervals for coculture infection experiments were calculated using the pooled standard error for ratios, as applied to the ratio between induced and uninduced conditions. Correlations are reported as Pearson linear regression coefficients. Boxplots follow the standard Tukey style and illustrate the 25th to 75th percentiles, with the median denoted by the center horizontal line and whiskers showing points within 1.5 times the interquartile range. Different experimental batches are shown as different shapes on plots where all data points are illustrated.

## Generation of cell lines and culture conditions

Human embryonic kidney (HEK-293T), lung adenocarcinoma (CaLu-3), and retinal pigment epithelium (hTERT RPE-1) cells were used in this study. HEK-293T and hTERT RPE-1 cells were grown in Dulbecco Modified Eagle Media (DMEM) supplemented with 10% fetal calf serum (FCS). Calu-3 cells were grown in Minimum Essential Medium (MEM) supplemented with 10% FCS, 2 mM GlutaMAX, 1 mM sodium pyruvate, and nonessential amino acids. All cell lines were maintained at 37˚C and 5% $CO_2$. To generate CaLu-3 CRISPRa cells, CaLu-3 cells were transduced with pHRSIN-$P_{SFFV}$-dCas9-VPR-$P_{SV40}$-Blast$^R$, and stable integrants were selected with Blasticidin.

## Vector cloning and lentiviral production

CRISPRa constructs and whole-genome guide library were created by the Weissmann lab and provided to us by the Tanenbaum lab. Individual CRISPRa guides were cloned into pCRIS-PRia-v2 (Addgene #84832). ACE2 and LRRC15 cDNAs were cloned into pHRSIN-cSGW vectors expressing puromycin, blasticidin, or hygromycin resistance cassettes driven by a pGK promoter. VSV-G lentiviruses were produced by transfection of HEK293T cells with a lentiviral expression vector and packaging vectors pCMVΔR8.91 and pMD.G at a DNA ratio of 3:2:1 using TransIT-293 (Mirus) following the manufacturer's recommendation.

## Production of SARS-CoV-2 viral stocks

For experiments using a fluorescent SARS-CoV-2 reporter virus encoding ZsGreen, the virus was encoded as previously described [46]. A total of 1 μg of pCCI-4K-SARS-CoV-2-ZsGreen plasmid DNA and 3 μL of Lipofectamine LTX with 3 μL of PLUS reagent in 100 μL optiMEM were used to transfect BHK-21 cells (ECACC) in 6-well plates. Supernatant was transferred to CaLu-3 cultures in 6-well plates, 72 hours post-transfection. Virus was allowed to propagate in CaLu-3 cells until clear cytopathic effect was observed, around 72 hours post-infection. Virus-containing supernatant was then utilized to seed larger cultures of CaLu-3 cells in T25 flasks to produce viral stocks utilized for infection assays. Viral titer was determined by 50% tissue culture infectious dose (TCID50) in CaLu-3 cells.

For coculture and viral adsorption experiments, the virus used was the lineage B viral isolate SARS-CoV-2/human/Liverpool/REMRQ0001/2020, provided by Ian Goodfellow (University of Cambridge), isolated early in the COVID-19 pandemic by Lance Turtle (University of Liverpool) and David Matthews and Andrew Davidson (University of Bristol) from a patient from the Diamond Princess cruise ship [18,90,91]. Viral stocks were prepared by passaging once in VeroE6 cells. In brief, cells were infected at a low MOI with the original viral stock and incubated for 72 hours (by which time cytopathic effect was evident). Viral titer was determined by TCID50 in VeroE6 cells.

## SARS-CoV-2 microscopy infection assays

RPE-1 or CaLu-3 cell lines were seeded into CellCarrier-96 Black optically clear bottom plates (Perkin Elmer) at a density of either approximately $1 \times 10^4$ cells/well 24 hours prior to infection or approximately $5 \times 10^4$ cells/well 72 hours prior to infection, respectively. Cells were infected with SARS-CoV-2-ZsGreen [46] at indicated MOIs and infection allowed to proceed for 48 hours followed by fixation by submerging plates in 4% formaldehyde for 15 minutes. Cells were stained with DAPI (Cell Signaling Technology, 0.1 μg/mL) at room temperature for 15 minutes prior to washing in PBS and imaging. Images were acquired using an ArrayScan XTI high-throughput screening microscope (Thermo Fisher) with a 10× magnification, using the 386 nm or 485 nm excitation/emission filter to detect DAPI and ZsGreen signal respectively across 16-fields per well. HCS Studio software (Thermo Fisher) was utilized to analyze images using the Target Activation application. Individual cells were identified by applying masks based on DAPI intensity, excluding small and large objects based on average nuclei size. The DAPI-generated nuclei masks were applied to the ZsGreen channel, providing a fluorescent signal intensity of SARS-CoV-2-ZsGreen detected in each cell. These data were exported as.fcs files and analyzed in FlowJo. Uninfected control wells were utilized to determine the fluorescence threshold to define SARS-CoV-2 infected ZsGreen$^+$ cells and the percentage of infected cells per well across all acquired data.

## Coculture infection experiments

To generate HEK293T cells expressing doxycycline-inducible LRRC15 (HEK293T-LRRC15), HEK293T cells were transfected with a PiggyBAC-based (PBQM812A-1, System Biosciences) doxycycline-inducible LRRC15 expression vector, together with the Super PiggyBac Transposase expression vector (PB210PA-1, System Biosciences) at a ratio of 2.5:1, then selected with 10 ug/mL blasticidin. To enrich for high levels of LRRC15 expression, cells were treated with 1 μg/mL doxycycline for 48 hours, stained with anti-LRRC15 antibody (ab150376, Abcam) and goat anti-rabbit secondary antibody conjugated to Alexa Fluor 647 (A21245, Thermo), then sorted for high Alexa Fluor 647.

Quantification of viral replication in A549 respiratory epithelial cells expressing ACE2 and a luciferase-based biosensor of SARS-CoV-2 infection (A549 targets) in coculture experiments was performed essentially as previously described [92,93]. In brief, 10,000 A549 targets were seeded in flat-bottomed 96-well plates and cocultured with the indicated ratios of control HEK293T cells or HEK293T cells expressing ACE2 (HEK293T-ACE2) or LRRC15 (HEK293T-LRRC15, either uninduced or induced with doxycycline) to achieve a total of 40,000 cells per well. The following morning, cells were infected with SARS-CoV-2 at MOI = 0.01. After 24 hours, cells were lysed in Bright-Glo Luciferase Buffer (Promega) diluted 1:1 with PBS and 1% NP-40. Lysates were transferred to opaque 96-well plates, and viral infection quantitated as Firefly luciferase activity measured using the Dual-Glo kit (Promega) according to the manufacturer's instructions. In these experiments, luciferase activity reflects the level of infection of A549 targets, and differences in luciferase activity between conditions reflect *trans*-acting effects of the cells included in coculture. To evaluate *trans*-acting effects of LRRC15 expression, luciferase activity was therefore compared between conditions in which A549 targets were cocultured with HEK293T-LRRC15 cells in the presence (LRRC15 high) or absence (LRRC15 low) of doxycycline.

## Viral adsorption experiments

For adsorption of viral particles, 40,000 HEK293T, HEK293T-ACE2, uninduced HEK293T-LRRC15, or Dox-induced HEK293T-LRRC15 cells were seeded in flat-bottomed 96-well

plates. The following morning, cells were incubated with the indicated amounts of SARS-CoV-2 viral stock for 4 hours (experiments conducted at different MOIs) or 1.5 hours (experiments conducted with multiple passages) in a total volume of 100 μL media. To quantify the infectious virus remaining post-adsorption, supernatants were transferred to HEK293T reporter cells expressing ACE2 and a luciferase-based biosensor of SARS-CoV-2 infection, and viral replication measured as luciferase activity at 24 hours as described above [92].

## Supporting information

**S1 Table. Ranked gRNA enrichment values for the genome-wide CRISPRa screen for gain of SARS-CoV-2 spike protein binding.** Enrichments were calculated using the RSA algorithm. CRISPRa, CRISPR activation; gRNA, guide RNA; RSA, redundant siRNA activity; SARS-CoV-2, Severe Acute Respiratory Syndrome Coronavirus 2.
(CSV)

**S1 Fig. Summary of raw data from arrayed cDNA screen.** Each well measured by flow cytometry contains cells transfected with an expression plasmid encoding full-length cDNA from our library encompassing the vast majority of human cell surface receptors. A majority of measured plates included a positive control well of ACE2 in the bottom right corner. As shown in main Fig 1C, hits from this screen aside from LRRC15 and already known spike receptors were found to be artifacts upon replication. ACE2, angiotensin-converting enzyme 2; LRRC15, leucine-rich repeat containing protein 15.
(TIF)

**S2 Fig. Flow cytometry validation of hits from CRISPRa screen.** An overview of binding signals upon replicating each sgRNA is shown (left) alongside western blots that validate one EBF1 sgRNA as slightly up-regulating ACE2 expression in RPE1 cells. Two exposures are shown, long (LE) and short (SE). The nearest molecular mass (in kilodaltons) of the standard is indicated next to each blot. ACE2, angiotensin-converting enzyme 2; CRISPRa, CRISPR activation.
(TIF)

**S3 Fig. SARS-CoV-2 spike protein retains binding to cells expressing LRRC15 when presented as trimers.** Representative flow cytometry traces depicting recombinant spike protein binding to HEK293 cells transfected to express the indicated receptors (left) are shown next to the quantified percentages of cells that bound spike. The y-axis is truncated at the indicated break point in order to display all conditions in a single plot. LRRC15, leucine-rich repeat containing protein 15; SARS-CoV-2, Severe Acute Respiratory Syndrome Coronavirus 2.
(TIF)

**S4 Fig. Cell-based spike binding affinity measurements for ACE2 receptor.** After incubating ACE2-transfected HEK293 cells with a range of monomeric spike protein concentrations, the binding saturation curve was measured to estimate an equilibrium dissociation constant. ACE2, angiotensin-converting enzyme 2.
(TIF)

**S5 Fig. CRISPRa sgRNA targeting *ACE2* or *LRRC15* in RPE-1 SunTag CRISPRa cell line specifically up-regulate transcription of the target gene.** qPCR measurements of mRNA for *ACE2* or *LRRC15* in cell lines transduced for the sgRNAs indicated along the x-axis. ACE2, angiotensin-converting enzyme 2; CRISPRa, CRISPR activation; LRRC15, leucine-rich repeat containing protein 15.
(TIF)

**S6 Fig. Deglycosylation of SARS-CoV-1 spike restores binding to LRRC15.** Comparison of SARS-CoV-1 spike to SARS-CoV-2 spike, where enzymatic removal of most N-linked glycans by PNGase F results in the SARS-CoV-1 spike gaining the ability to bind LRRC15 at similar levels to SARS-CoV-2 spike. LRRC15, leucine-rich repeat containing protein 15; SARS-CoV-1, Severe Acute Respiratory Syndrome Coronavirus 1; SARS-CoV-2, Severe Acute Respiratory Syndrome Coronavirus 2.
(TIF)

**S7 Fig. Localization of LRRC15 exogenously expressed in CaLu-3 cells.** Two fields of view are shown for CaLu-3 cells stained at different dilutions of monoclonal anti- human LRRC15 antibody. LRRC15 was predominantly detected along the cell plasma membrane, with some possible faint staining in intracellular compartments. LRRC15, leucine-rich repeat containing protein 15.
(TIF)

**S8 Fig. Coculturing ACE2-expressing A549 respiratory epithelial cells with LRRC15-expressing HEK293T cells does not appreciably change rates of infection.** (**A**) Schematic of the experiments. The A549 target cells are respiratory epithelium that express ACE2 and a luciferase-based biosensor of SARS-CoV-2 infection. HEK293T cells express doxycycline-inducible LRRC15. (**B**) Surface LRRC15 expression can be readily induced by doxycycline with minimal background staining in the uninduced state. Flow cytometry traces show HEK293T cells in different expression conditions (red for constitutive ACE2 expression, light blue for uninduced LRRC15, blue for induced LRRC15 expression, and grey for unmodified HEK293T cells). Representative data from at least 4 independent experiments. These data are also referenced in S9B Fig. (**C**) Coculture of A549 target cells with HEK293T cells expressing ACE2 significantly reduces rates of infection by authentic SARS-CoV-2 virus. A549 respiratory epithelial cells expressing ACE2 and a luciferase-based biosensor of SARS-CoV-2 infection (A549 targets) were mixed with different ratios of control HEK293T cells or HEK293T cells expressing ACE2 (HEK293T-ACE2), then infected with SARS-CoV-2 (MOI = 0.01). Infection of A549 targets is quantitated as the fold-change in luciferase activity in infected versus uninfected cells. Mean values ± SD are shown for an experiment performed in triplicate, representative of 2 independent experiments. (**D**) Coculture of A549 target cells with HEK293T cells expressing LRRC15 does not reduce rates of infection by authentic SARS-CoV-2 virus. A549 respiratory epithelial cells expressing ACE2 and a luciferase-based biosensor of SARS-CoV-2 infection (A549 targets) were mixed with different ratios of control HEK293T cells or HEK293T cells expressing doxycycline-inducible LRRC15 (HEK293T-LRRC15) in the presence or absence of doxycycline, then infected with SARS-CoV-2 (MOI = 0.01). Infection of A549 targets is quantified as the ratio of luciferase activity in conditions with (LRRC15 high) or without (LRRC15 low) doxycycline induction, with a ratio of 1 implying no effect of LRRC15 expression. Bars show the pooled standard error around the mean value from experiments done in triplicate, representative of two independent experiments. Statistical significance was tested using one-way ANOVA followed by Dunnett's test for multiple comparisons. ****$p < 0.0001$. ACE2, angiotensin-converting enzyme 2; LRRC15, leucine-rich repeat containing protein 15; SARS-CoV-2, Severe Acute Respiratory Syndrome Coronavirus 2.
(TIF)

**S9 Fig. SARS-CoV-2 preadsorption by LRRC15-expressing HEK293T cells does not appreciably change rates of infection.** (**A**) Schematic of the experiments. HEK293T cells express ACE2 or doxycycline-inducible LRRC15. Reporter cells express ACE2 and a luciferase-based biosensor of SARS-CoV-2 infection. (**B**) Surface LRRC15 expression can be readily induced by

doxycycline with minimal background staining in the uninduced state. Flow cytometry traces show HEK293T cells in different expression conditions (red for constitutive ACE2 expression, light blue for uninduced LRRC15, blue for induced LRRC15 expression, and grey for unmodified HEK293T cells). Representative data from at least 4 independent experiments. These data are also referenced in S8B Fig. (**C**) Preadsorption of authentic SARS-CoV-2 virus by HEK293T cells expressing ACE2 but not LRRC15 significantly reduces rates of infection across a range of MOIs. HEK293T, HEK293T-ACE2, uninduced HEK293T-LRRC15, or Dox-induced HEK293T-LRRC15 cells were incubated with the indicated volume of SARS-CoV-2 viral stock for 4 hours to allow adsorption of viral particles. Each 0.1 μL of viral stock corresponds to MOI ≈ 0.02. Supernatants were then transferred to reporter cells expressing ACE2 and a luciferase-based biosensor of SARS-CoV-2 infection. Infection of reporter cells is quantified as % maximum luminescence at 24 hours. Mean values ± SEM are shown for an experiment performed in triplicate, representative of 4 independent experiments. (**D**) Multiple rounds of preadsorption of authentic SARS-CoV-2 virus by HEK293T cells expressing ACE2 but not LRRC15 enhances the reduction in infection. HEK293T, HEK293T-ACE2, uninduced HEK293T-LRRC15, or Dox-induced HEK293T-LRRC15 were incubated with SARS-CoV-2 viral stock (MOI = 0.01) for 1.5 hours to allow viral adsorption (passage 1). A sample of each supernatant from passage 1 was stored at 4°C, and the remainder subject to a further round of preadsorption for 1.5 hours (passage 2). This step was repeated twice (total 4 passages). All supernatants were then transferred to reporter cells expressing ACE2 and a luciferase-based biosensor of SARS-CoV-2 infection. Infection of reporter cells is quantified as % maximum luminescence at 24 hours. Mean values ± SEM are shown for an experiment performed in triplicate, representative of 2 independent experiments. Statistical significance was tested using two-way ANOVA followed by Dunnett's test for multiple comparisons. $^{*}p < 0.05$, $^{***}p < 0.001$, $^{****}p < 0.0001$. ACE2, angiotensin-converting enzyme 2; LRRC15, leucine-rich repeat containing protein 15; SARS-CoV-2, Severe Acute Respiratory Syndrome Coronavirus 2.
(TIF)

**S10 Fig. Phylogeny of LRRC15 protein sequence conservation across proposed SARS-CoV-2 hosts and related mammalian species.** The percentages of LRRC15 residues that are identical to human are shown at each terminal branch. LRRC15, leucine-rich repeat containing protein 15; SARS-CoV-2, Severe Acute Respiratory Syndrome Coronavirus 2.
(TIF)

**S11 Fig. Example gating strategy for cell line flow cytometry measurements of recombinant protein binding.** The hierarchy of gates goes from left to right.
(TIF)

**S1 Data. Numerical values corresponding to S1 Fig.**
(XLSX)

**S2 Data. Numerical values corresponding to Fig 1C.**
(XLSX)

**S3 Data. Numerical values corresponding to Fig 1D.**
(XLSX)

**S4 Data. Numerical values corresponding to S2 Fig.**
(XLSX)

**S5 Data. Numerical values corresponding to Fig 1E.**
(XLSX)

**S6 Data. Numerical values corresponding to Fig 1F.**
(XLSX)

**S7 Data. Numerical values corresponding to Fig 2A.**
(XLSX)

**S8 Data. Numerical values corresponding to S3 Fig.**
(XLSX)

**S9 Data. Numerical values corresponding to Fig 2B.**
(XLSX)

**S10 Data. Numerical values corresponding to Fig 2C.**
(XLSX)

**S11 Data. Numerical values corresponding to Fig 2D.**
(XLSX)

**S12 Data. Numerical values corresponding to S4 Fig.**
(XLSX)

**S13 Data. Numerical values corresponding to Fig 2E.**
(XLSX)

**S14 Data. Numerical values corresponding to S5 Fig.**
(XLSX)

**S15 Data. Numerical values corresponding to Fig 3B.**
(XLSX)

**S16 Data. Numerical values corresponding to S6 Fig.**
(XLSX)

**S17 Data. Numerical values corresponding to Fig 3C.**
(XLSX)

**S18 Data. Numerical values corresponding to Fig 3D.**
(XLSX)

**S19 Data. Numerical values corresponding to Fig 3E.**
(XLSX)

**S20 Data. Numerical values corresponding to Fig 3F.**
(XLSX)

**S21 Data. Numerical values corresponding to Fig 4A.**
(XLSX)

**S22 Data. Numerical values corresponding to Fig 4B.**
(XLSX)

**S23 Data. Numerical values corresponding to Fig 4C.**
(XLSX)

**S24 Data. Numerical values corresponding to Fig 4D.**
(XLSX)

**S25 Data. Numerical values corresponding to Fig 4E.**
(XLSX)

**S26 Data. Numerical values corresponding to Fig 4F.**
(XLSX)

**S27 Data. Numerical values corresponding to Fig 4G.**
(XLSX)

**S28 Data. Numerical values corresponding to Fig 4H.**
(XLSX)

**S29 Data. Numerical values corresponding to S8 Fig.**
(XLSX)

**S30 Data. Numerical values corresponding to S9 Fig.**
(XLSX)

**S31 Data. Numerical values corresponding to S10 Fig.**
(XLSX)

**S1 Raw Images. Uncropped protein gel and western blot images.**
(PDF)

## Acknowledgments

An RPE-1 cell line expressing dCas9-SunTag$_{10x\_v4}$-P2A-mCherry and scFv-GCN4-GFP-VP64 was provided by Marvin Tanenbaum and Jonathan Weissman. pCCI-4K-SARS-CoV-2-ZsGreen was a gift from Sam Wilson, University of Glasgow. The viral isolate SARS-CoV-2/human/Liverpool/REMRQ0001/2020 was a gift from Ian Goodfellow, University of Cambridge. FACS experiments were enabled by Dr. Anna Petrunkina Harrison, and Arrayscan experiments were enabled by Veronika Romashova at the JCBC FACS core facilities. Thanks to Liane Dupont and Zheng-Shan Chong for useful advice on the design of CRISPRa screens.

## Author Contributions

**Conceptualization:** Jarrod Shilts, Thomas W. M. Crozier, Ana Teixeira-Silva, Paul J. Lehner, Gavin J. Wright.

**Formal analysis:** Jarrod Shilts, Thomas W. M. Crozier, Ana Teixeira-Silva.

**Funding acquisition:** Nicholas J. Matheson, Paul J. Lehner, Gavin J. Wright.

**Investigation:** Jarrod Shilts, Thomas W. M. Crozier, Ana Teixeira-Silva, Ildar Gabaev, Pehuén Pereyra Gerber, Edward J. D. Greenwood, Samuel James Watson, Brian M. Ortmann, Christian M. Gawden-Bone, Tekle Pauzaite.

**Methodology:** Jarrod Shilts, Thomas W. M. Crozier, Ana Teixeira-Silva, Ildar Gabaev, Pehuén Pereyra Gerber, Edward J. D. Greenwood, Samuel James Watson, Brian M. Ortmann, Christian M. Gawden-Bone, Tekle Pauzaite.

**Project administration:** James A. Nathan, Paul J. Lehner, Gavin J. Wright.

**Resources:** Markus Hoffmann, Stefan Pöhlmann.

**Supervision:** James A. Nathan, Stefan Pöhlmann, Nicholas J. Matheson, Paul J. Lehner, Gavin J. Wright.

**Validation:** Jarrod Shilts, Thomas W. M. Crozier, Ana Teixeira-Silva.

**Visualization:** Jarrod Shilts, Thomas W. M. Crozier.

**Writing – original draft:** Jarrod Shilts.

**Writing – review & editing:** Jarrod Shilts, Thomas W. M. Crozier, Ana Teixeira-Silva, Ildar Gabaev, Pehuén Pereyra Gerber, Edward J. D. Greenwood, Samuel James Watson, Brian M. Ortmann, Christian M. Gawden-Bone, Tekle Pauzaite, Markus Hoffmann, James A. Nathan, Stefan Pöhlmann, Nicholas J. Matheson, Paul J. Lehner, Gavin J. Wright.

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
