## [Editor Report · Decision Letter 0]

18 Oct 2021

Dear Dr. Shilts, 

Thank you for submitting your manuscript entitled "LRRC15 mediates an accessory interaction with the SARS-CoV-2 spike protein" for consideration as a Short Reports by PLOS Biology.

Your manuscript has now been evaluated by the PLOS Biology editorial staff, as well as by an academic editor with relevant expertise, and I am writing to let you know that we would like to send your submission out for external peer review.

Once your full submission is complete, your paper will undergo a series of checks in preparation for peer review. Once your manuscript has passed the checks it will be sent out for review. 

If your manuscript has been previously reviewed at another journal, PLOS Biology is willing to work with those reviews in order to avoid re-starting the process. Submission of the previous reviews is entirely optional and our ability to use them effectively will depend on the willingness of the previous journal to confirm the content of the reports and share the reviewer identities. Please note that we reserve the right to invite additional reviewers if we consider that additional/independent reviewers are needed, although we aim to avoid this as far as possible. In our experience, working with previous reviews does save time. 

If you would like to send your previous reviewer reports to us, please specify this in the cover letter, mentioning the name of the previous journal and the manuscript ID the study was given, and include a point-by-point response to reviewers that details how you have or plan to address the reviewers' concerns. Please contact me at the email that can be found below my signature if you have questions. 

Please re-submit your manuscript within two working days, i.e. by Oct 20 2021 11:59PM.

Kind regards,

Paula

Paula Jauregui, PhD

Associate Editor

PLOS Biology

---

## [Decision Letter · Decision Letter 1]

7 Dec 2021

Dear Dr. Shilts,

Thank you for submitting your manuscript "LRRC15 mediates an accessory interaction with the SARS-CoV-2 spike protein" for consideration as a Short Reports at PLOS Biology. Your manuscript has been evaluated by the PLOS Biology editors, an Academic Editor with relevant expertise, and by several independent reviewers.

In light of the reviews (below), we will not be able to accept the current version of the manuscript, but we would welcome re-submission of a much-revised version that takes into account the reviewers' comments. We cannot make any decision about publication until we have seen the revised manuscript and your response to the reviewers' comments. Your revised manuscript is also likely to be sent for further evaluation by the reviewers.

You will see that all the reviewers agree and have similar concerns. They all find the work novel and interesting but they all have issues that should be solved before publication. In particular, you should use the whole virus, test LRRC15 and ACE2 co-expression, and provide more rigorous data about enhancement of viral infection.

We expect to receive your revised manuscript within 3 months. 

**IMPORTANT - SUBMITTING YOUR REVISION**

*Re-submission Checklist*

*Published Peer Review*

*PLOS Data Policy*

*Blot and Gel Data Policy*

Sincerely,

Paula

---

Paula Jauregui, PhD

Associate Editor

PLOS Biology

REVIEWS:

Reviewer #1: Teunis B.H. Geijtenbeek. Lectin type and other virus receptors.

Reviewer #2: Coronavirus entry.

Reviewer #3: Receptor-ligand pathways.

Reviewer #1: This study has identified LRRC15 as a protein that interacts with the Spike protein from SARS-CoV-2. Two independent screens have been performed and identified LRRC15. Further studies showed that LRRC15 interacts at the CTD of S protein and overlapping with ACE2 binding site. The authors suggest that LRRC15 is expressed in the same tissues but overlap with ACE2 expression is not very strong when analysing singe cell transcriptome data sets from public domain. Expression of LRRC15 did not render cells susceptible to SARS_CoV-2 recombinant virus. The authors suggest that LRRC15 might enhance infection as ectopic expression of LRRC15 in Calu-3 cells increases infection at low concentrations. 

Identification of LRRC15 as a protein that interacts with S protein from SARS-CoV-2 is interesting and novel. Binding data with soluble S proteins and overexpression LRRC15 are convincing, although binding is weak compared to ACE2. Using public datasets the authors suggest that there is some overlap with ACE2 in the lung but these data are not very convincing. Functional relevance to SARS-CoV-2 remains unclear; LRRC15 overexpression did not allow infection by a recombinant SARS-CoV-2 in absence of ACE2. Based on ectopic expression in Calu-3 cells the authors suggest that it might enhance infection but this conclusion is not fully supported by the infection data. Finally, as the authors have not shown any convincing co-expression of LRRC15 with ACE2, it remains unclear what the relevance is of ectopic expression of LRRC15. Could LRRC15 be induced by inflammation or stimuli?

Major concerns

- Most data are from binding assays using soluble S protein. Fig 1E and Fig 2A show that LRRC15 interacts with S protein but very weakly. The authors suggest that this is due to low expression LRRC15 as binding is comparable to antibody staining (Fig 2A). The latter cannot be concluded as techniques are different (binding vs antibody binding, dimers etc). The authors could perform binding with actual SARS-CoV-2 particles to show that LRRC15 interacts with complete virus containing endogenous S protein. This would support the binding data obtained with S protein. 

- The authors compare binding with other known receptors for SARS-CoV-2 and conclude that LRCC15 does not affect or induce binding to these receptors (line 161, fig 2). It is interesting to compare other receptors but it does not prove that LRRC15 binding is direct. Binding studies with recombinant LRRC15 and S protein (in ELISA for example) would support such a conclusion. This would also support the weak binding data observed (see above).

- The authors state that the binding to LRRC15 is specific for SARS-CoV-2 but this is not correct and should be changed as suppl fig 5 shows that removal of glycans allows binding of LRRC15 with S protein from SARS-CoV-1. These data are very interesting and should be in main text, moreover, it suggest that the cell-lines and expression system might affect the binding of SARS-CoV-1 and SARS-CoV-2 as cell-lines have different glycosylation biases. Therefore would be interesting if the S protein from SARS-CoV-1 and -2 is overexpressed by more relevant cells such as lung epithelial cell-lines. 

- The expression data using public data sets are overstated, even though there is a overlap in tissues between ACE2 and LRRC15, there is no overlap in lung cells. These data do not provide any relevant information. It would be more important to show whether ACE2 positive cells such as alveolar macrophages or epithelial cells (primary or cell-lines) express also LRRC15. The authors use standard cultured Calu-3 cells but these are very difficult to infect with SARS-CoV-2 due to low ACE2. Calu-3 cells need to be differentiated in monolayers to induce strong expression of ACE2 and expression of LRRC15 could be analysed in such a model. Even if there is no co-expression, LRRC15 could facilitate binding and transfer to other cells. But the co-expression needs to be more carefully investigated. 

- Infection data using a recombinant SARS-CoV-2 is interesting and suggest that LRRC15 is not a receptor but could enhance infection. However, the enhanced infection of LRRC15-positive ACE2 cells is not very clear and lost when using higher MOI. Blocking reagents to LRRC15 (antibodies) would be useful to determine whether it is indeed the LRRC15 function that enhances infection. Here Calu-3 cells are not grown in a monolayer which affects infection. These experiments should be done on with differentiated Calu-3 cells in a monolayer as these are more susceptible to SARS-CoV-2 (and might already express LRRC15 but at the very least it is more relevant to SARS-CoV-2 infection). 

Wild-type SARS-CoV-2 is highly infectious and infection experiments using SARS-CoV-2 isolates would support the observed data. 

minor comment

line 226-228. The sentence 'These findings suggest a novel function of the receptor-binding region of the SARS-CoV-2 spike protein which appears to be a relatively recent evolutionary innovation' is vague and not supported by the data. 

Reviewer #2: PBIOLOGY-D-21-02640 review

This submission demonstrates that SARS-CoV-2 spike protein ectodomain fragments bind a protein called LRRC15. This is a new discovery that adds one more spike-binding factor to several others that have been previously reported. The discovery of LRRC15 came from two well-executed genetic screens, and the subsequent documentation of LRRC15 as a spike-binding protein was convincingly demonstrated. The paper is strong for its thorough controls. However, the project and its impact is weakened with results that do not convincingly demonstrate a role for LRRC15 in virus infection. 

1. Results in Fig 4E are used to claim a proviral role for LRRC15, but these are amongst the weakest data of the report. LRRC15 overexpression very modestly increases percentage of SARS2-positive Calu3 cells and only when virus is input at low input moi. At higher input moi, LRRC15 overexpression appears to have no effect. Also, increasing moi generated fewer infected cells overall. What explains this unusual inverse correlation between input moi and cell infection? Also, Fig 4E expt and data could be expanded to determine whether LRRC15 accelerates infection, or could be expanded to other cell types beyond Calu3. In sum here, if the title of the paper remains as "LRRC15 mediates an "accessory" interaction"; and if the discussion keeps lines 326-327, that "LRRC15 modulates a host cells susceptibility to SARS-CoV-2" then further support of these claims, beyond Fig 4E, is recommended.

2. Binding data: The results include a great deal of binding data but do not include any data on the binding of complete virus particles to LRRC15. This seems like a major omission, particularly because spike trimers on virus particles have different conformation and glycosylation patterns than S1 or uncleaved S1-S2 ectodomains produced independent of virus infection (and given that glycosylation patterns appear to be central to LRRC15 binding, i.e., suppl fig 5). 

3. Coincidence of LRRC15 and ACE2 on target cells: Data show that only small cell proportions bind spikes, and this seems to hold for both LRRC15 binding and ACE2 binding. Are spike binding potentials of LRRC15 and ACE2 on the same small populations? If not, then what is the model for how LRRC15 is proviral?

4. The discussion section makes several intriguing points about LRRC15 but (at least for this reviewer) does not put a clear view forward on LRRC15 activity in infection. One could envision abundant low-affinity binding agents capturing viruses at the same RBD sites that ACE2 uses for entry activation, and then presenting other presumably unbound viral spikes to ACE2 in some way, but this and alternative views are not provided. There are opportunities in the discussion to clarify how the LRRC15 might operate in infection, or, if the LRRC15 does not operate in infection but rather captures viruses "unproductively" then that could be communicated. 

Reviewer #3: In this manuscript, Dr. Wright and Lehner's groups have used their cutting-edge cell-based systemic screens to determine novel SARS-CoV-2 spike binding partners. They used ACE2 as a positive control for their screens, and identified LRRC15 as a new interaction partner specifically for the spike protein of SARS2, but not SARS1. A serial of delicate experiments were also performed to verify this interaction, particularly focusing on its difference with heparan sulfates or lectin receptors. The binding epitope was mapped to the c-terminus of S1. Functionally, similar to some of the other recently reported SARS2 interacting partners (lectins, LRP1 etc.), LRRC15 itself does not permit viral entry but may enhance ACE2-dependent SARS-2 infection as indicated by a reporter virus. Given the fact that LRRC15 is expressed in human lung vasculature cells and fibroblast, this interaction may have some implication in the physiological condition during SARS2 infection. 

Overall, this finding is novel and the study was well-performed, particularly the systemic screens led by leaders in the field. The paper was also well-written. However, there are several technical concerns I see for potential improvement.

1. Other than ACE2, there are several S interaction partners have been recently identified (Several C-type lectins, LRP1, AXL, TLR4 etc). However, in the current screens, only CLEC4M was used as positive control beside ACE2. Did both screens contain these reported spike binding partners? This is important as these information will validate the robustness of the screening systems described in this paper.

2. The LRRC15 binding data look quite weak in general, even much weaker than CLEC4M (Fig 1C), which may because of 1) it is a weak interaction or 2) the insufficient expression of LRRC15 on the cell-surface (Fig 2A for instance). The authors may need to establish a LRRC15 high-expression cell line and then test the binding. Alternatively, are there any LRRC15 positive cancer cell lines can be used for this purpose? Moreover, the authors used a cell-based titration system to monitor the KD and found it was around 260nM. However, this system is not accurate. The authors need to at least test the direct protein-protein interaction of Spike/LRRC15 ECD by ELISA or Octet to calculate the more reliable KD.

3. The HSPG and EDTA data is quite interesting, which indicate that LRRC15 binding is likely different from HSPG/Lectin binding of SARS2 spike. Does this interaction glycan dependent, or involve any glycosylations sites in the spike? The authors claimed that this interaction differs from previously described spike-binding receptors, however, which is quite over-stated. The authors only mapped this interaction to the CTD of Spike, how about RBD (major binding interface for ACE2 binding) and the rest of the CTD (which are mostly named as CTD)? The CTD region is quite big, and the authors need more detail mapping to locate the exact binding region of LRRC15. Moreover, ACE2 competition data may have other interpretation beyond sharing the same epitope---such as steric hindrance. 

4. As LRRC15 does not support SARS-2 infection, however, the authors found that LRRC-15 overexpression in Calu-3 cells slightly increase SARS-reporter virus infection. This data is important but need some controls--for example, did LRRC15 affect ACE2 expression? How strong the expression of LRRC15 can lead to more SARS-2 infection in this system? Moreover, given that LRRC15 is expressed in lung vascular/lymphatic endothelium or fibroblast cells, what is the percentage of LRRC15+ cells that are ACE2 positive (and vice versa)? It seems that the chance of co-expression of ACE2 to LRRC15 is quite low, which may affect the interpretation of the importance of this study, particularly in the physiological condition.

5. C-type lectins have been suggested to support SARS2 transinfection, how about LRRC15?

---

## [Decision Letter · Decision Letter 2]

15 Jun 2022

Dear Dr. Shilts,

Thank you for your patience while your revised manuscript "LRRC15 mediates an accessory interaction with the SARS-CoV-2 spike protein" was being assessed at PLOS Biology. Your manuscript has been evaluated by the PLOS Biology editors, an Academic Editor with relevant expertise, and the same reviewers that saw the original work.

As you will see in the reviewer reports, which can be found at the end of this email, some reviewers are not satisfied with the revision. Based on their specific comments and following discussion with the Academic Editor, we consider that we cannot publish the manuscript in its current form. However, given our and the reviewer interest in your study, we would be open to inviting a comprehensive revision of the study that thoroughly addresses all the reviewers' comments.

In particular, given that the scope of this Short Report is to show the discovery and characterization of the binding

interaction between LRRC15 and the SARS-CoV-2 spike protein, we consider that it is important to show in the manuscript S-LRRC15 interaction by using either SARS-CoV-2 or a pseudotyped virus, if that is technically easier. If using virus to show the interaction is not possible, it is important that you show the interaction with either recombinant protein or overexpressing the proteins in physiologically relevant cells. One of the main claims of the manuscript is that LRRC15 acts in cis, therefore, it would be important to show co-expression of ACE2 and LRRC15 in relevant cells. Reviewer #1 suggests another approach for the experiments to see whether LRRC15 acts in trans, this will definitely be interesting and will add to the manuscript but it will not be necessary for publication. You might want to consider adding some of the data from the revision into the manuscript, as reviewer #3 suggests. 

We remain editorially interested in the story, but that these issues were raised before and addressing them would be required for publication. Please, let us know if you feel you will not be able to address them.

Given the extent of revision that would be needed, we cannot make a decision about publication until we have seen the revised manuscript and your response to the reviewers' comments. Your revised manuscript would need to be seen by the reviewers again, but please note that we would not engage them unless their main concerns have been addressed. 

We would like to receive the revision in 3 months. Please email us (plosbiology@plos.org) if you have any questions or concerns, or envision needing a (short) extension.

**IMPORTANT - SUBMITTING YOUR REVISION**

*Resubmission Checklist*

*Published Peer Review*

*PLOS Data Policy*

*Blot and Gel Data Policy*

Sincerely,

Paula

---

Senior Editor

PLOS Biology

REVIEWS:

Reviewer's Responses to Questions

PLOS authors have the option to publish the peer review history of their article (what does this mean?). If published, this will include your full peer review and any attached files.

Reviewer #1: Yes: Teunis BH Geijtenbeek

Reviewer #2: No

Reviewer #3: No

Reviewer #1: the authors have not addressed my important concerns.

It is important to analyse the co-expression of LRRC15 and ACE2. The addition of data on lung fibroblasts during inflammation is interesting but does not address the concern that RCC15 cannot affect ACE2 positive cells or enhance infection.

major concerns

1. no data are provided that SARS-CoV-2 virus binds to LRRC15 (my initial question). The authors refer to other preprints but that is insufficient proof.

2. no data are provided that S binds recombinant LRCC15. there is a discussion about why it does not work and referred to a preprint, but this is not sufficient. the authors should show that S protein binds LRCC15.

3. no data are provided on the expression of S protein in more relevant cell-lines. some attempts have been made but no results have been obtained. 

4. no primary virus is used and this decreases the relevance and impact. 

5. trans infection. the data presented in R10 are not clear and do not really allow assessment of trans. a easier setup would be to incubate LRRC15 cells with SARS-CoV-2, wash after incubation period and add to ACE2 positive cells. this allows assessment of transmission and is unaffected by co-culture of cells. The experiments have not been performed well and therefore this needs to be revisited. the preprints showing transmission have used different more successful setup.

Reviewer #2: The authors have made appropriate revisions in response to input from the reviewers. Of note, the authors provided very extensive responses to every one of the reviewers' concerns, and they performed additional experiments. While not all experiments yielded useful findings, some did and they have been incorporated into the current version. Also notably, there are now three independent discoveries of LRRC15 as a SARS-CoV-2 binding ligand, and one report links LRRC15 to COVID19 outcomes. The findings in this submission mostly cohere with these other reports, and in those cases where they do not, the authors have offered some explanations for the perceived discordances. All in all, the authors communicated their findings clearly and professionally. Given that several groups have landed on LRRC15 as a SARS-CoV-2 binding ligand of potential significance to virus biology and COVID19, this paper is considered timely and highly significant to the field. 

Reviewer #3: The authors have performed amazing amount of work to answer my previous concerns. I greatly appreciate their extensive efforts and was really impressed by their careful dissection of these comments. The major issue now is the LRRC15 biology upon infection. Given the fact that two other papers suggested LRRC15 is inhibitory to SARS-CoV-2 in trans, it would be hard to evaluate the current data from this paper that LRRC15 modestly promotes infection in cis with ACE2. This observation is also not physiological as very little ACE2+ cells have LRRC15 expression. Nevertheless, this paper is still rather helpful to the research community. In this regard, it would be helpful to include into the paper the negative data from the authors that LRRC15 in trans could not inhibit SARS-CoV-2 real virus infection through ACE2, and potentially the validation data of other Spike receptors, Figure R5, which will raise important discussions to the field.

---

## [Editor Report · Decision Letter 3]

29 Nov 2022

Dear Dr. Shilts,

Thank you for your patience while we considered your revised manuscript "LRRC15 mediates an accessory interaction with the SARS-CoV-2 spike protein" for publication as a Short Reports at PLOS Biology. This revised version of your manuscript has been evaluated by the PLOS Biology editors and the Academic Editor.

Based on our Academic Editor's assessment of your revision, we are likely to accept this manuscript for publication, provided you satisfactorily address the following data and other policy-related requests.

1. DATA POLICY:

A) Supplementary files (e.g., excel). Please ensure that all data files are uploaded as 'Supporting Information' and are invariably referred to (in the manuscript, figure legends, and the Description field when uploading your files) using the following format verbatim: S1 Data, S2 Data, etc. Multiple panels of a single or even several figures can be included as multiple sheets in one excel file that is saved using exactly the following convention: S1_Data.xlsx (using an underscore).

B) Deposition in a publicly available repository. Please also provide the accession code or a reviewer link so that we may view your data before publication.

Regardless of the method selected, please ensure that you provide the individual numerical values that underlie the summary data displayed in the following figure panels as they are essential for readers to assess your analysis and to reproduce it: Figures 1CDEF, 2ABCDE, 3BCDEF, 4ABCDEFGH, and supplementary figures SF1, SF2, SF3, SF4, SF5, SF6, SF8BCD, SF9BCD, SF10.

**Please also ensure that figure legends in your manuscript include information on where the underlying data can be found, and ensure your supplemental data file/s has a legend.**

We require the original, uncropped and minimally adjusted images supporting all blot and gel results reported in an article's figures or Supporting Information files. We will require these files before a manuscript can be accepted so please prepare and upload them now.

Please provide this for figures 2F, 3A, and supplementary figure SF2.

Please carefully read our guidelines for how to prepare and upload this data: https://journals.plos.org/plosbiology/s/figures#loc-blot-and-gel-reporting-requirements.

We expect to receive your revised manuscript within two weeks.

*Published Peer Review History*

*Press*

Sincerely,

Paula

---

Senior Editor,

pjaureguionieva@plos.org,

PLOS Biology

---

## [Editor Report · Decision Letter 4]

14 Dec 2022

Dear Dr. Shilts,

Thank you for the submission of your revised Short Reports "LRRC15 mediates an accessory interaction with the SARS-CoV-2 spike protein" for publication in PLOS Biology. On behalf of my colleagues and the Academic Editor, Ken Cadwell, I am pleased to say that we can in principle accept your manuscript for publication, provided you address any remaining formatting and reporting issues. These will be detailed in an email you should receive within 2-3 business days from our colleagues in the journal operations team; no action is required from you until then. Please note that we will not be able to formally accept your manuscript and schedule it for publication until you have completed any requested changes.

PRESS

Sincerely, 

Paula 

---

Senior Editor

PLOS Biology
